# Bile acid-dependent transcription factors and chromatin accessibility determine regional heterogeneity of intestinal antimicrobial peptides

Yue Wang [1,10], Yanbo Yu[1,2,3,10], Lixiang Li[1,2,3,10], Mengqi Zheng[1,10], Jiawei Zhou[1], Haifan Gong[4], Bingcheng Feng[1], Xiao Wang[5], Xuanlin Meng[6], Yanyan Cui[7], Yanan Xia[1], Shuzheng Chu[7], Lin Lin[1], Huijun Chang[1], Ruchen Zhou[1], Mingjun Ma[1], Zhen Li[1,2,3], Rui Ji[1,2,3], Ming Lu [8], Xiaoyun Yang[1,2,3], Xiuli Zuo [1,2,3] ✉, Shiyang Li [1,7,9] ✉ & Yanqing Li [1,2,3] ✉

Antimicrobial peptides (AMPs) are important mediators of intestinal immune surveillance. However, the regional heterogeneity of AMPs and its regulatory mechanisms remain obscure. Here, we clarified the regional heterogeneity of intestinal AMPs at the single-cell level, and revealed a cross-lineages AMP regulation mechanism that bile acid dependent transcription factors (BATFs), *NR1H4*, *NR1H3* and *VDR*, regulate AMPs through a ligand-independent manner. Bile acids regulate AMPs by perturbing cell differentiation rather than activating BATFs signaling. Chromatin accessibility determines the potential of BATFs to regulate AMPs at the pre-transcriptional level, thus shaping the regional heterogeneity of AMPs. The BATFs-AMPs axis also participates in the establishment of intestinal antimicrobial barriers of fetuses and the defects of antibacterial ability during Crohn's disease. Overall, BATFs and chromatin accessibility play essential roles in shaping the regional heterogeneity of AMPs at pre- and postnatal stages, as well as in maintenance of antimicrobial immunity during homeostasis and disease.

The antimicrobial peptides (AMPs) play a critical role in host protection by directly killing or hindering the growth of microorganisms[1,2]. Paneth cells enriched in the ileum were considered as the predominant cell type for the expression of intestinal AMPs[3,4]. However, novel AMP-expressing cells in different intestinal regions, such as human colonic

*BEST4*+ enterocytes and goblet cells expressing *LYPD8* and *WFDC2*, respectively[5], and mouse ileal goblet cells expressing *Defa24*[6], have been identified in recent studies, suggesting potential anatomical regional heterogeneity of AMPs. Previous studies have elucidated the signaling pathways regulating Paneth cell differentiation[7–10] and

[1]Department of Gastroenterology, Qilu Hospital of Shandong University, Jinan, China. [2]Laboratory of Translational Gastroenterology, Qilu Hospital of Shandong University, Jinan, China. [3]Shandong Provincial Clinical Research Center for digestive disease, Jinan, China. [4]School of Computer Science and Engineering, Sun Yat-sen University, Guangzhou, China. [5]Department of Pathology, Qilu Hospital of Shandong University, Jinan, China. [6]State Key Laboratory of Microbial Metabolism, Joint International Research Laboratory of Metabolic and Developmental Sciences, and School of Life Sciences and Biotechnology, Shanghai Jiao Tong University, Shanghai, China. [7]Advanced Medical Research Institute, Shandong University, Jinan, China. [8]Clinical Epidemiology Unit, Qilu Hospital of Shandong University, Jinan, China. [9]Key Laboratory for Experimental Teratology of Ministry of Education, Shandong University, Jinan, China. [10]These authors contributed equally: Yue Wang, Yanbo Yu, Lixiang Li, Mengqi Zheng. ✉e-mail: zuoxiuli@sdu.edu.cn; lishiyang@sdu.edu.cn; liyanqing@sdu.edu.cn

revealed the roles of microenvironmental signals such as microorganism, vitamin D, and bile acids (BAs) in regulating the expression of AMPs in Paneth cells but have failed to elucidate the mechanisms generating the regional heterogeneity of intestinal AMPs[11–14].

Although transcriptional profiles of single cells have been used to map the cell atlas of fetal gut and identify α-defensin-expressing mature Paneth cells at the prenatal ileum[15,16], when and how the fetal gut establishes antimicrobial barrier against inflammatory damage caused by the first colonization of the flora at birth within a prenatal environment lacking microbes[17–19], which are thought to be a critical factor in the induction of AMP production has not been clarified[2,11,14]. In addition, striking Paneth cell deficiency and disordered AMPs production occur in the ileum during Crohn's disease (CD)[4], but the regulatory mechanisms remain obscure.

Here, we combine single-cell RNA sequencing (scRNA-seq), single-cell assay for transposase-accessible chromatin (scATAC-seq), chromatin immunoprecipitation followed by sequencing (ChIP-seq), conditional gene knockouts and intestinal organoids to reveal the regional heterogeneity of AMPs in human and mouse intestines, and find an antimicrobial program expressed in multiple cell lineages in the ileum. Our data demonstrate a regulation of AMPs that bile acid dependent transcription factors (BATFs), *NR1H4*, *NR1H3*, and *VDR*, regulate the

expression of AMPs in a ligand-independent manner (i.e. BATFs-AMPs axis). The chromatin accessibility determines the potential of BATFs to regulate AMPs expression at the pre-transcriptional level, thus shaping the regional heterogeneity of AMPs between the small intestine (SI) and large intestine (LI). In addition, our analysis reveal that the BATFs-AMPs axis is involved in establishing intestinal antibacterial barrier during fetal development, and its disorder leads to the dysfunction of the antibacterial function of multiple lineages during CD.

## Results

### AMP-mediated regional immune surveillance in the intestine
To explore the regional heterogeneity of AMPs and their alterations during disease and development, we established an intestinal cell landscape by integrating our newly generated scRNA-seq data of human duodenum, jejunum, ileum tissues (14 samples) and eight previously published sc(sn)RNA-seq and scATAC-seq datasets (Supplementary Data 1)[15,16,20–24]. It includes 371 intestinal biopsy samples and over 780,000 cells across species, anatomical regions, developmental time (organoids, pre- and postnatal), and health and disease states (Fig. 1a, b). According to the transcriptional characteristics, unsupervised clustering preliminarily divided the cells into six compartments, including epithelium, T/innate lymphoid cells (ILCs), B/

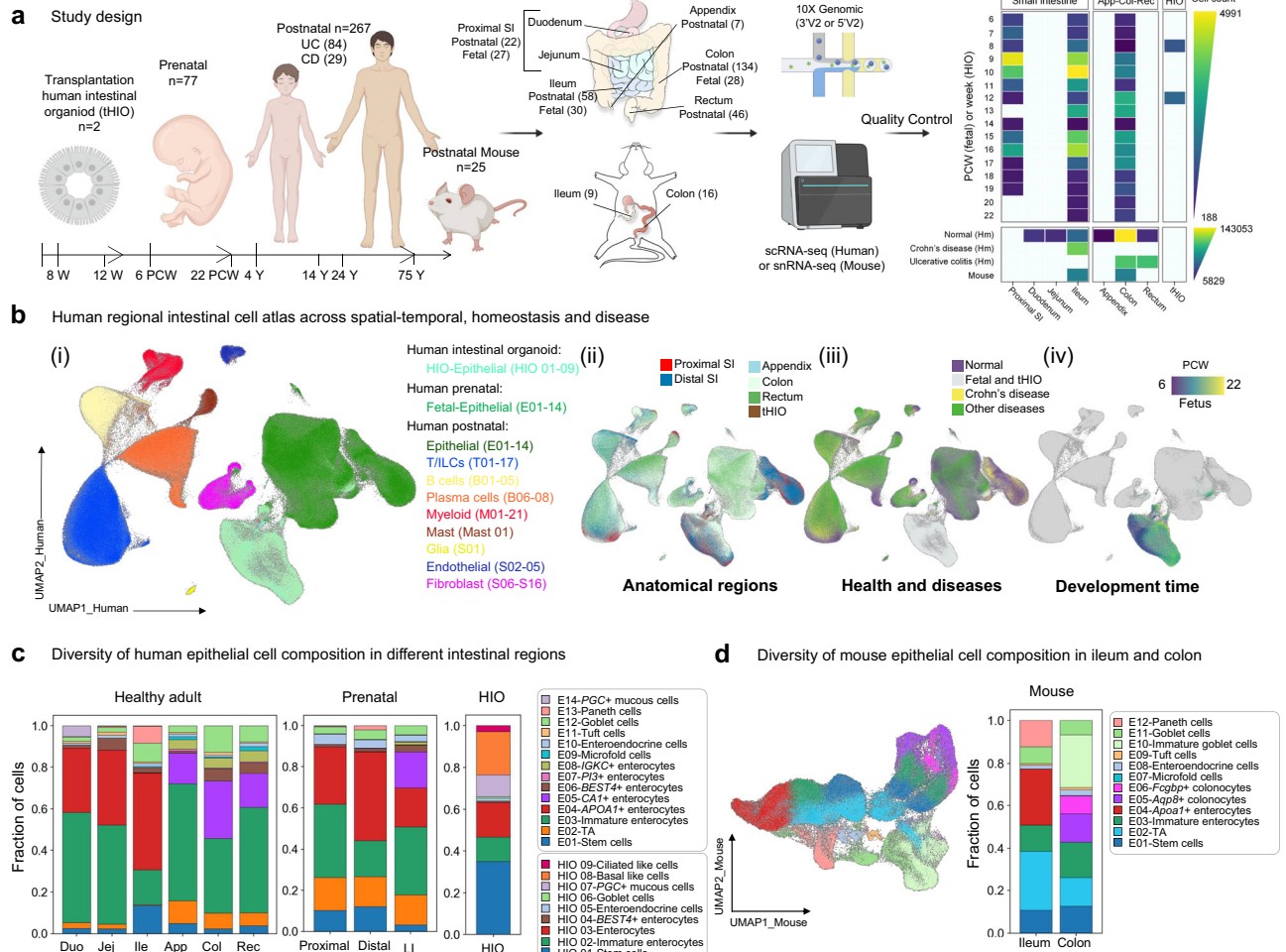

**Fig. 1 | Intestinal cellular atlas across species, anatomical regions, developmental time, and health and disease states. a** Summary of the scRNA-seq data that were integrated in this work (left) and scRNA-seq experiment phenotype overview matrix-plot depicting phenotype across species, anatomical regions, developmental time, diseases and high-quality post-QC cells recovered per phenotype. Created with BioRender.com. **b** UMAP (uniform manifold approximation and projection) embedding by major cell partitions (i), anatomical regions (ii), healthy states (iii), or development time (iv). **c** Diversity of human epithelial cell composition in different intestinal regions at healthy adult (left), prenatal stage (middle), or human intestinal organoid (right). **d** UMAP embedding (left) and cellular composition of mouse intestinal epithelium in different regions (right). Source data underlying Fig. 1a, c and d are provided as a Source Data file.

plasma cells, stromal cells, mononuclear phagocytes (MNPs), and mast cells (Fig. 1b). Based on marker genes, 80 subsets were identified, of which, a total of 314,722 and 127,856 strictly quality-controlled cells were categorized into 16 human and 12 mouse epithelial cell types, respectively (Fig. 1c, d and Supplementary Fig. 1a–c, 2a–d).

On this basis, the AMP landscape was mapped for different intestinal anatomical regions in healthy humans (duodenum, jejunum, ileum, appendix, colon, rectum) and mice (ileum, and colon) during the postnatal period (Fig. 2a, b). The results showed that 40 AMPs were highly expressed in human intestinal epithelial cells with significant region-specificity (Fig. 2c; Supplementary Fig. 3a–d and 4a). The mouse data further suggest that the regional heterogeneity of AMPs is evolutionarily conserved but that the specific antimicrobial peptide gene types differ significantly across species (Fig. 2d). According to their anatomical sites of high expression, the AMPs can be classified as SI-specific AMPs, which are represented by α-defensins (*DEFA5, DEFA6/Defa20, Defa24*); C-type lectins (*REG3A, REG3G/Reg3b, Reg3g*); proteinases which regulate the activity of defensins (*PRSS2/Mmp7*); large intestine (LI)-specific AMPs, which are represented by *WFDC2/Wfdc2;*

and SI and colorectal co-expressed AMPs, such as *LCN2* and *PLA2G2A* (the mouse gene symbols follow the "/" symbols) (Fig. 2c, d; Supplementary Fig. 4a, b).

Notably, our data suggested that most SI-specific AMPs (i.e. *DEFA5/6, REG3A/G, PRSS2,* and *LYZ*) except *ITLN2*, were not restricted to Paneth cells in the SI (Fig. 2c, d and Supplementary Fig. 3a, b, d). *LGR5*+ stem cells and transit-amplifying (TA) cells in the human SI also expressed high levels of SI-specific AMPs (*DEFA5/6, REG3A/G, PRSS2,* and *LYZ*) (Supplementary Fig. 3a, b, d, e and 4c–e). Due to the difference in the number of *DEFA5*+ stem cells versus Paneth cells, stem cells serve as an important source of α-defensins, *REG3A* and *REG3G* in the human proximal SI, besides Paneth cells (Fig. 2e, f and Supplementary Fig. 3a). To verify the above findings, we employed RNAScope in situ hybridization experiments for *LGR5* and *DEFA5* (Fig. 2g and Supplementary Fig. 4d), which, together with the co-staining immunofluorescence images of LGR5 and DEFA5 (Fig. 2h), demonstrate that not only is DEFA5 mRNA transcribed in SI stem cells, but DEFA5 protein is also translated. Additionally, we found out the reason why previous scRNA-seq studies defined Paneth cells as the sole source of SI-specific

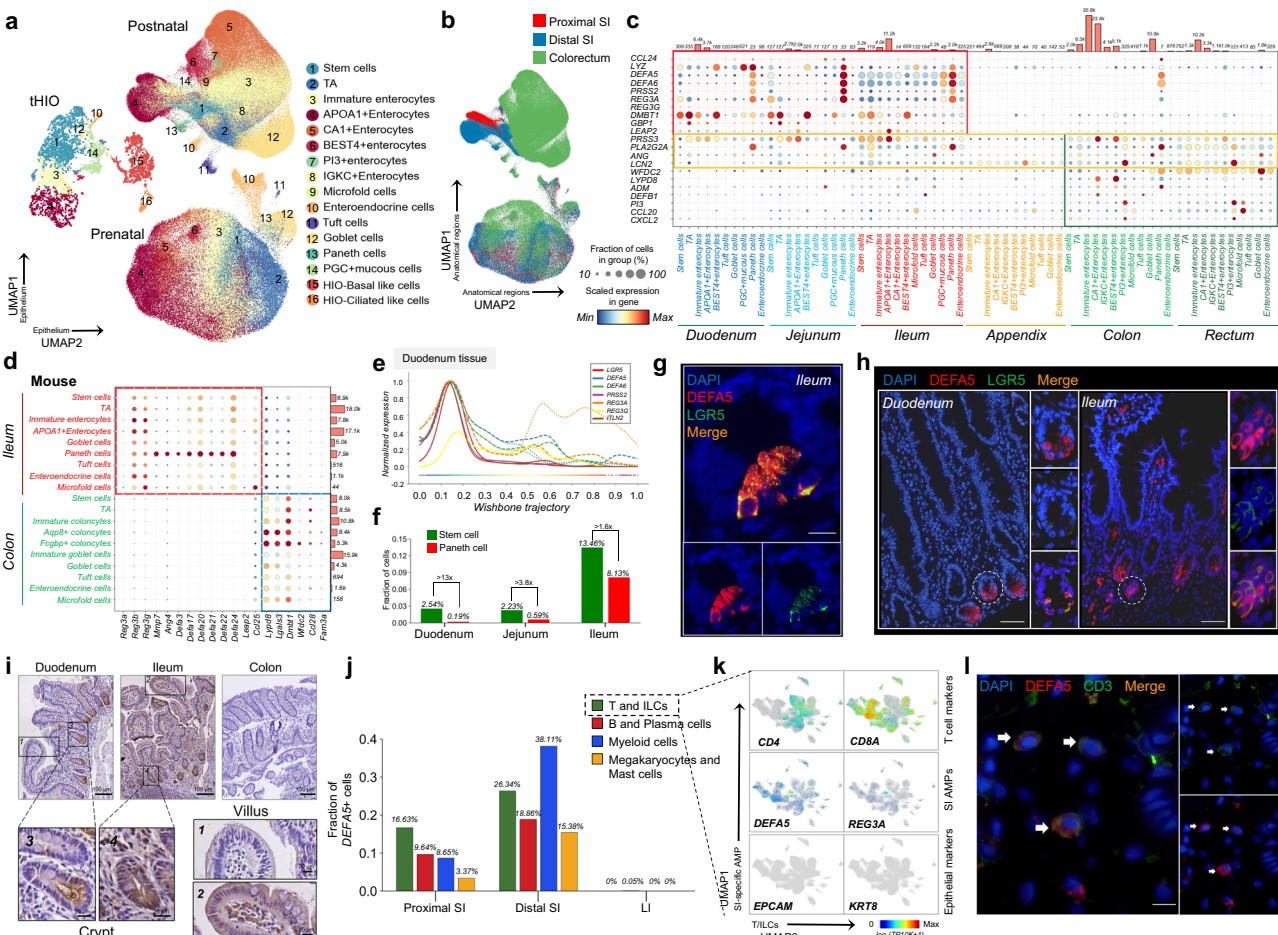

**Fig. 2 | AMP-mediated regional immune surveillance in the intestine.**
**a** Combined visualization of three UMAP embeddings of prenatal and postnatal human intestinal epithelium and human organoids. **b** UMAP embedding overlay showing the regional subsets of pre- and postnatal epithelial cells. **c, d** Dot plot showing the regional and cellular specificity of AMPs in human (**c**) and mouse (**d**). **e** The trajectory dendrogram of duodenum epithelial cells highlighting the expression of SI-specific AMPs (*DEFA5, DEFA6, PRSS2, REG3A, REG3G, ITLN2*) correlates positively with *LGR5*. **f** Bar plot depicting the proportion of stem cells and Paneth cells in the epithelium of duodenum, jejunum and ileum in scRNA-seq data. **g** RNAScope in situ hybridization of *DEFA5* (red), *LGR5* (green), and DAPI (blue) in bowel sections from the human ileum. Scale bars, 10 μm. Staining repeated on two

participants. **h** Bowel sections from the human intestine were immunofluorescent stained for DEFA5 (red), LGR5 (green), and DAPI (blue). Scale bars, 50 μm. Staining repeated on three participants. **i** Expression of DEFA5 in histological sections (*n* = 3 for each region). Scale bars, 100 or 20 μm. **j** Bar plot depicting the proportion of *DEFA5*+ cells from T/ILCs, B/Plasma, Myeloid, Megakaryocytes/Mast cell compartments are different between proximal SI, distal SI, and LI. **k** Feature plots showing expression of T cell markers, SI-specific AMPs, and epithelial markers in T/ILCs compartment. **l** Bowel sections from the human ileum were immunofluorescent stained for DEFA5 (red), CD3 (green), and DAPI (blue). Arrows point to T cells expressing DEFA5. Scale bars, 10 μm. Staining repeated on two participants. Source data underlying Fig. 2f and j are provided as a Source Data file.

AMPs, such as *DEFA5/6* and *REG3A*[15]. This is because the fetal epithelial data and postnatal LI epithelial data they used masked SI-specific AMPs expression in non-Panth epithelial cells in the postnatal ileum (Supplementary Fig. 5a, b).

Surprisingly, *LYZ* was widely expressed in the epithelial cells of proximal SI (Supplementary Fig. 3d), resulting in a higher level of *LYZ* than in the ileum even with fewer Paneth cells (Supplementary Fig. 3f and Fig. 2f). To validate this, we analyzed a published RNA-seq dataset (E-MTAB-1733)[25] and compared the *LYZ* expression in human duodenum and ileum. The results indicate that, consistent with the scRNA-seq data, *LYZ* has a significantly higher expression level in the duodenum than in the ileum (Supplementary Fig. 3g).

Furthermore, based on regional heterogeneity, the cell type specificity of AMPs was characterized; for example, *BEST4*+ enterocytes in the duodenum and jejunum specifically expressed *NPY*[5,15,22], Tuft cells in the colon and rectum specifically expressed *CAMP*, and goblet cells in the SI specifically expressed *CCL24* (Supplementary Fig. 4a). Collectively, regional heterogeneity and cell type specificity of intestinal AMPs at the single-cell level were identified, which may shape microbial communities in different regions of the gut.

## Regional milieu determines program of AMPs co-expressed in multiple lineages

We noted that human and mouse ileal epithelium ubiquitously expressed SI-specific AMPs in various epithelial lineages, including absorptive and secretory ones (Fig. 2c, d, i). Furthermore, we were surprised to find that SI-specific AMPs, such as *DEFA5*, *DEFA6* and *REG3A* were widely expressed by even immune cells in the SI, especially in the distal SI (Fig. 2j, k and Supplementary Fig. 6a–f). A strict doublet detection procedure, which removed all droplets detected to express the epithelial cell markers *EPCAM* and *KRT8*, was performed for the T/ILCs and B/Plasma cell compartments to exclude the effect of doublets of epithelial versus immune cells. The remaining small intestinal T and B cells still expressed high levels of SI-specific AMPs (Fig. 2k and Supplementary Fig. 6d). To verify the above findings, we analyzed the expression of *DEFA5* in immune cells of 44 postnatal human organs using a external validation scRNA-seq dataset[26]. The results indicated that, consistent with our internal discovery dataset, *DEFA5* was highly expressed in the immune cells of SI and was barely expressed in other tissues (Supplementary Fig. 6g–k). In addition, we also confirmed the expression of DEFA5 in ileal CD3D + T cells at the protein level by immunofluorescence (Fig. 2l).

These results suggest that the regional program of AMPs is possibly due to specific environmental signals enriched in the ileum on various lineage cells, rather than the lineage-specific effect by promoting the differentiation of specific cell types (such as Paneth cells).

## BATFs are the upstream regulators of intestinal AMPs

To explore the mechanism of regional heterogeneity of AMPs, we established transcription factor fate decision trees for epithelial cells spanning different developmental periods (organoid, fetal development, child, adult) and anatomical regions (proximal SI, distal SI, colorectum)[27]. We found that the BATFs, *NR1H4, NR1H3, NR1H2, and NR1I3*, are key regulons modulating the differentiation of regional epithelial cell subsets and are pivotal switches regulating cell fate toward SI regions, especially the distal SI (Supplementary Fig. 7a and Supplementary Data 2).

In addition, to investigate non lineage-specific upstream regulators of AMPs, we analyzed differentially expressed genes (DEGs) of absorptive enterocytes from the proximal, distal SI and colorectum as well as differentially activated pathways, and predicted upstream regulators of DEGs using ingenuity pathway analysis (IPA)[28]. The results showed that enterocytes of the distal SI highly expressed many SI-specific AMPs and were significantly enriched in AMP and BAs processing pathways (Fig. 3a–c). IPA further indicated multiple BATFs and

BAs to be upstream regulators of DEGs in the distal SI (Fig. 3d and Supplementary Data 3). We therefore focused our attention on probing whether BAs and BATFs regulate the AMPs.

We constructed a TF-target gene regulatory network for AMPs (Supplementary Fig. 7b) based on TF-target gene expression correlations reflected by scRNA-seq data and a priori DNA sequence knowledge of gene promoter regions and TF motifs[27], identifying that the BATFs, *NR1H4, NR1H3, VDR, and NR1I3*, were upstream regulators of 15, 9, 8, and 1 AMP genes, respectively (Fig. 3e). It is worth noting that *NR1H4* regulates most of the known Paneth cell-AMPs, including *LYZ, DEFA5, DEFA6, PRSS2, REG3A*, and *PLA2G2A*. Consistently, highly expressed *NR1H4* in multiple ileal epithelial cells may account for the ubiquitous expression of SI AMPs (Fig. 2c, i and Fig. 3b). Furthermore, Paneth cells, which express the highest level of SI-specific AMPs, are also the lineage with the highest expression of BATFs in SI secretory epithelial cell types (Fig. 3f and Supplementary Fig. 7c). Therefore, we reasoned a cross-lineage mechanism of AMP regulation that BATFs, may act as upstream TFs, directly regulate the expression of downstream AMP target genes (i.e. BATFs-AMPs axis).

## BATFs act as transcription factors to directly regulate the expression of AMPs

Conditional knockouts were employed to validate the proposed mechanism (BATFs-AMPs axis). To investigate the in vivo effects of BATFs deletion, we compared the transcript levels of AMPs in the intestinal mucosa of *Nr1h4* floxed mice (control) with intestinal epithelial-specific knockout (*Nr1h4*-intKO), liver-specific KO (*Nr1h4*-livKO), and systemic KO (*Nr1h4*-totKO) mice (Supplementary Fig. 8a)[29]. We also investigated the transcript levels of AMPs in the distal ileal and colonic mucosa of *Vdr*-KO/Tg transgenic mice that only express *VDR* in the distal intestine (distal ileum, cecum and colorectum) (control) versus *Vdr*-intKO mice (Supplementary Fig. 8b)[30].

We observed a striking downregulation of AMPs in intestinal epithelium specific BATFs knockout mouse mucosal tissue as expected (Supplementary Fig. 8c, d). *Vdr*-intKO mice showed significant decreases in the SI-specific AMPs *Reg3b* and *Reg3g* and no significant alteration in *Wfdc2* compared to *Vdr*-KO/Tg mice (Fig. 3g). Moreover, *Nr1h4*-intKO mice exhibited significant downregulation of the SI-specific AMPs *Lyz1, Defa24, Reg3b* and *Reg3g* and the LI-specific AMPs *Wfdc2* relative to *Nr1h4*-floxed mice (Fig. 3h and Supplementary Fig. 8c). ChIP-seq data further demonstrated, within the gut tissue, BATFs FXR, LXR (aliases for *NR1H3* gene) and VDR binding at the promoter regions of AMP genes (Fig. 3i, j and Supplementary Fig. 7d–f). These results suggest that BATFs are directly involved in the regulation of AMPs transcription, which may be controlled by the expression levels of BATFs in different intestinal regions.

## BATFs regulate the expression of AMPs in a ligand-independent manner

Considering that the BATFs have both ligand-dependent and ligand-independent regulatory capacities[31–33], we further explored whether the regulation of AMPs by BATFs requires direct binding of ligands. We first analyzed transcriptome data from intestinal tissues of mice fed with specific pharmacologic agonists for FXR (PX20606), LXR (GW3965), and VDR (1,25(OH)2D3) (Fig. 4a, top)[30,34,35]. The results showed that these agonists enhanced the expression of known downstream genes (i.e., *Fabp6*, *Abcg5*, and *Cyp24a1*) in epithelial cells (Fig. 4b)[30,34,35], but did not cause significant upregulation of AMPs (Fig. 4c–e), suggesting that BATFs regulate AMPs in a ligand-independent manner.

Whether and how BAs are involved in the regulation of AMPs is currently a controversial issue. Chenodeoxycholic acid (CDCA) and cholic acid (CA), which are endogenous agonist of FXR, upregulate Paneth cell-AMPs, *Defa5*, *Defa20* and *Defa23*, in murine ileal explants[13]. However, another study in which a Western diet resulted in Paneth cell

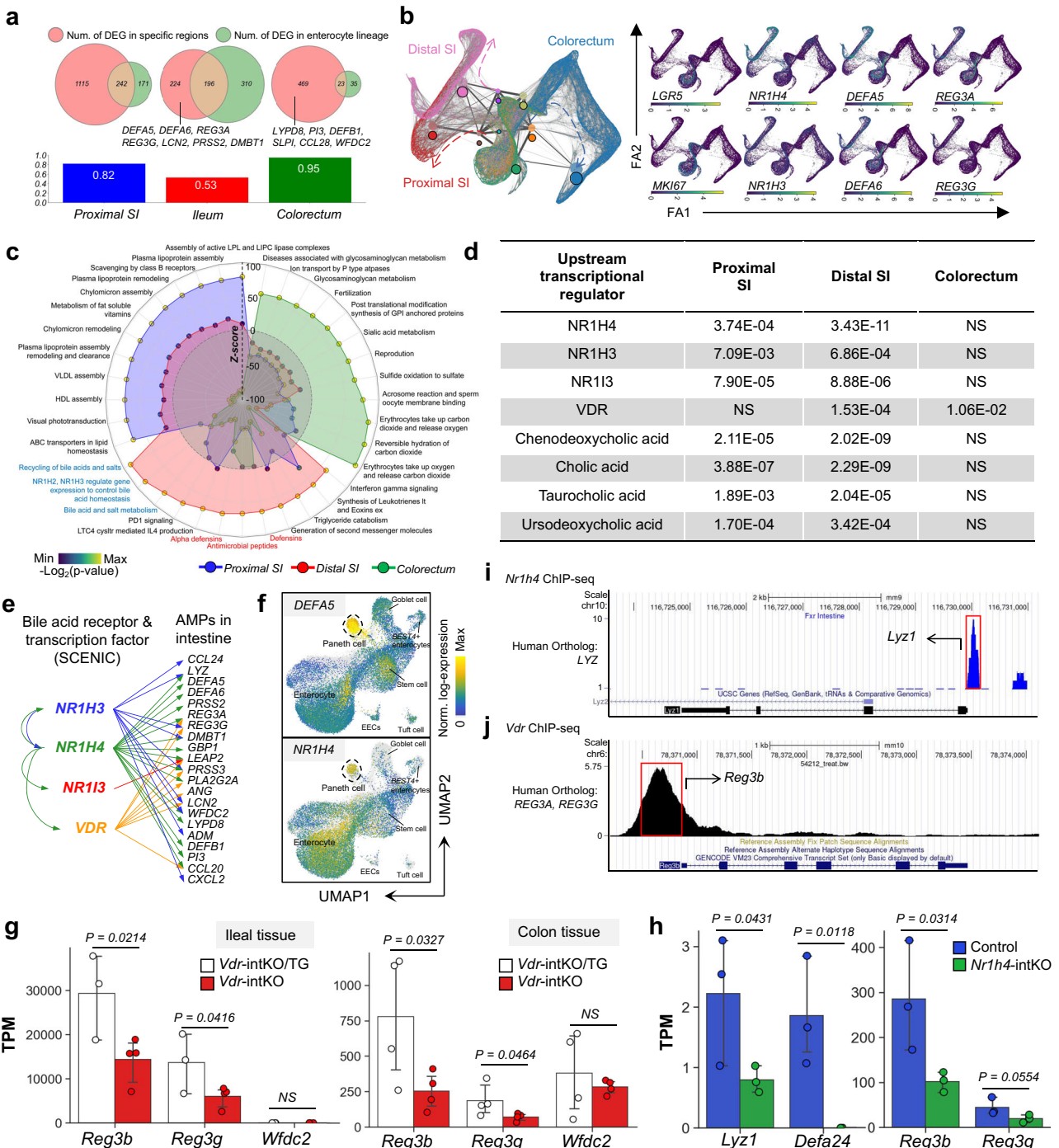

**Fig. 3 | BATFs are the upstream regulators of intestinal AMPs. a** Expression of region-specific AMPs is a common feature of all cell lineages in specific intestinal region. Left circle (red) represents the differential genes of absorptive enterocytes (*APOA1*+ and *CA1* + ) in specific region from absorptive enterocytes in the other two regions; Right circle (green) represents the differential genes between absorptive enterocytes and other epithelial subsets within this region. Bar plots show the proportion of un-overlapped part of left circles. **b** Trajectory and partition-based graph abstraction of regional epithelial cells maps the differentiation of regional subsets (left) and their marker genes (right). **c** Pathway analysis of absorptive enterocytes. Color in the circles reflects *p*-value. **d** Ligand-dependent nuclear receptors and chemicals (bile acids) depicted by IPA of genes enriched in absorptive enterocytes in a specific region versus absorptive enterocytes in the other two regions. *P*-values were determined by IPA for genes in differential expression analysis. **e** Visualization of the regulatory relationship between BATFs and AMPs predicted by pySCENIC. **f** Feature plots showing mRNA expression of *DEFA5* and *NR1H4* in small intestinal epithelial cells. **g** Bulk RNA-seq comparing AMPs expression between *Vdr*-KO/Tg mice (Control, *n* = 3 biologically independent samples for ileum, *n* = 4 biologically independent samples for colon) and *Vdr*-KO mice (*n* = 4 biologically independent samples for ileum and colon) in ileum (left) and colon (right). **h** Bulk RNA-seq comparing AMPs expression of *Nr1h4* floxed mice (control) with *Nr1h4*-intKO mice in colon tissue (*n* = 3 biologically independent samples for each group). **i, j** Small intestinal ChIP-seq analysis of NR1H4 (**i**) and VDR (**j**) binding sites in AMP genes. The length of the error bars is a 95% confidence interval for the mean in Fig. 3. NS not significant. All *p*-values were calculated and reported using one-tailed Student's *t*-test. Source data underlying Fig. 3g, h are provided as a Source Data file.

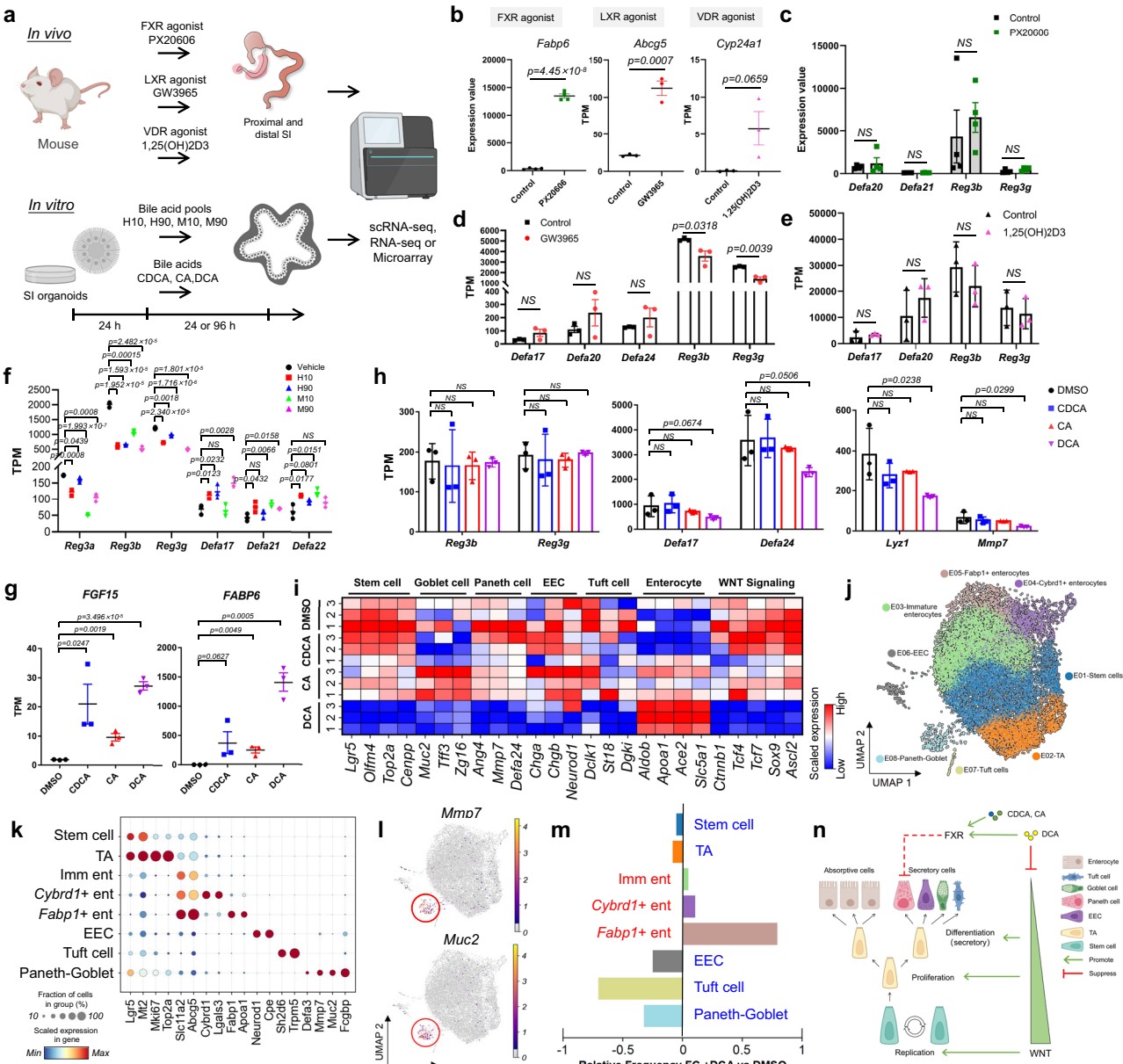

**Fig. 4 | BATFs regulate the expression of AMPs in a ligand-independent manner. a** Schematic of in vivo and in vitro experiments. Created with BioRender.com. **b** Shown are the expression of known downstream genes of BATFs after feeding with FXR (left), LXR (middle), and VDR (right) agonists or vehicles. Three (LXR and VDR agonists) or four (FXR agnoist) representative replicates are shown. **c**–**e** Shown are the expression of predicted downstream SI-specific AMPs of BATFs after feeding with FXR (**c**), LXR (**d**), and VDR (**e**) agonists or vehicles. Three (**d**, **e**) or four (**c**) representative replicates are shown. **f** Shown are the expression of predicted downstream SI-specific AMPs of BATFs in mouse small intestinal organoids treated with vehicle or 4 distinct bile acid pools, which varied based on the species (i.e., human (H) or mouse (M)), and the proportion of 12a-hydroxylated bile acids (10% or 90%). Three representative replicates are shown. **g** Shown are the expression of known downstream genes (*FGF15* and *FABP6*) of FXR in mouse small intestinal organoids treated with bile acids or DMSO. Three representative replicates are shown. **h** Shown are the expression of predicted downstream SI-specific AMPs of

BATFs in mouse small intestinal organoids treated with CDCA, CA, DCA, or vehicle (i.e., DMSO). Three representative replicates are shown. **i** Heapmap of changes in epithelial cell markers (Stem cell, Goblet cell, Paneth cell, EEC, Tuft, Enterocyte) and WNT signaling in mouse small intestinal organoids treated with bile acids or DMSO. Three representative replicates are shown. **j** UMAP embedding of 8 epithelial cell subsets in mouse SI organoids (*n* = 13,815 cells). **k** Dot plot showing the relative expression and the fraction of cells expressing selected markers across epithelial clusters. Two representative markers for each cluster are plotted. **l** Feature plots showing the expression of marker genes of in vivo Paneth cell (*Mmp7*) and Goblet cell (*Muc2*) in organoid cells. **m** Bar plot showing the relative fold change of cell populations in organoids treated with or without DCA. **n** Schematic of the mechanism of bile acids regulate the expression of AMPs. Created with BioRender.com. All data are mean ± SEM. NS not significant. The *p*-values of Fig. 4b–f and Fig. 4g, h were calculated using two-tailed Student's *t*-test and one-tailed Student's *t*-test respectively. Source data underlying Fig. 4b–i are provided as a Source Data file.

defects reported seemingly contradictory results, with deoxycholic acid (DCA) inhibiting ileal Paneth cell function through the FXR signaling[36].

To further demonstrate that BATFs regulate AMPs in a ligand-independent manner and to clarify mechanistic insights into BAs

regulate the expression of AMPs, we utilized an organoid model and RNA-seq to probe for altered gene expression in organoids under different BAs stimuli (Fig. 4a, bottom). The results showed that compositionally different BA mixtures led to varying degrees of altered AMPs in mouse SI organoids (Fig. 4f), suggesting that the BAs are

indeed involved in the regulation of AMPs. Then, we analyzed transcriptome changes in SI organoids stimulated with CDCA, CA, and DCA and showed that the three BAs all significantly upregulated known FXR downstream genes (*Fgf15* and *Fabp6*) (Fig. 4g)[34,37]. However, only DCA led to a strong downregulation of Paneth cell-AMPs, such as *Defa17*, *Defa24*, *Mmp7*, and *Ang4* (Fig. 4h), suggesting that DCA does not cause Paneth cell defects through FXR signaling. In addition, we did not observe significant upregulation of Paneth cell-AMPs by CDCA and CA (Fig. 4h).

## DCA suppresses SI-specific AMPs by inducing Paneth cell differentiation defects

To further explore the mechanism by which DCA causes Paneth cell deficiency, we analyzed the markers of absorptive and secretory epithelial lineages in the mouse ileum (Supplementary Fig. 1c) to be altered upon stimulation with different BAs. The results showed that DCA downregulated the markers of stem cell and all secretory lineages including tuft cell, goblet cell, Paneth cell, and enteroendocrine cell (EEC), whereas significantly upregulated markers of the absorptive enterocyte (Fig. 4i), indicating that DCA may affected the direction of Paneth cell differentiation. ScRNA-seq experiments in mouse intestinal organoids stimulated with DCA and DMSO were employed to directly depict the changes in the proportion of epithelial lineages. In 13,815 quality-controlled cells, we identified 8 cell subpopulations, including 2 undifferentiated lineages (Stem cell, TA), 3 absorptive enterocytes (Immature enterocyte, *Fabp1*+ enterocyte, *Crbrd1*+ enterocyte), and 3 secretory epithelial lineages (Tuft cell, EECs, Paneth-Goblet cell) (Fig. 4j). Notably, the specific markers of adult mouse Paneth cells, *Mmp7* and *Defa3*, were co-expressed with Goblet cell markers *Muc2* and *Fcgbp* in the same cells of the organoids (Fig. 4k, l), which is consistent with the findings of a previous scRNA-seq study on mouse small intestinal organoids[38]. The changes in the proportions of each cell type after DCA stimulation indicated that DCA induced a decrease in the proportions of stem cells and secretory epithelial cells, and an increase in the proportion of absorptive enterocytes (Fig. 4m), confirming that DCA induced Paneth cell differentiation defects rather than suppressing the expression of Paneth cell-AMPs.

Considering that Wnt signaling promotes stem cell replication and TA cell proliferation, as well as promoting TA cell differentiation into secretory lineages[39–41], we analyzed transcript levels of the primary genes involved in Wnt signaling and showed that DCA strongly downregulated *Ctnnb1*, *Tcf4*, *Tcf7*, and *Sox9* (Fig. 4i). In conclusion, our results clarified the effect of the BAs on AMPs. CDCA and CA did not cause significant changes in the expression of AMPs in epithelial cells. DCA suppresses the differentiation of secretory lineages by inhibiting the WNT signaling pathway, leading to the down-regulation of Paneth cell-AMPs, rather than through FXR signaling (Fig. 4n). In addition, these results further confirmed that BATFs regulate AMPs in a ligand-independent manner.

## Chromatin accessibility determines the potential of BATFs to regulate AMPs in different gut regions

The proposed BATFs-AMPs axis can explain that the SI-specific AMPs in the ileum are higher than those in the proximal SI according to the difference in the abundance of BATFs. However, there are still 3 questions that cannot be explained. First, why do the BATFs predicted to be upstream of both SI-specific and LI-specific AMPs regulate only one kind of AMPs in a specific intestinal region (Figs. 2c, 3e). For example, the predicted downstream of *NR1H4* includes both SI-specific AMP *REG3A* and LI-specific AMP *WFDC2*, but only *REG3A* is expressed in the SI (Figs. 2c, 3e). Second, why does the liver tissue express high BATFs not express intestinal AMPs (Fig. 5a, b). Third, why ileal enterocytes express *NR1H4* at similar levels to Paneth cells but relatively low levels of SI-specific AMPs (Fig. 3f).

To address the above questions, we analyzed the binding ability of BATFs to the promoter regions of AMP target genes within mouse liver and SI tissues using *Nr1h4* (FXR) as an example. ChIP-seq of FXR showed that although liver tissue highly expressed *Nr1h4*, the binding level of FXR to AMP target genes was significantly lower within the liver than the SI (Fig. 5c and Supplementary Fig. 9a), suggesting a pre-transcriptional level of mechanism controls the potential of BATFs to regulate the expression of AMPs within different organs.

We therefore used scATAC-seq to establish a single-cell chromatin accessibility map of liver, ileal and colonic epithelial cells (Fig. 5d). We observed significant differences in the chromatin accessibility of intestinal AMP genes including SI-specific AMPs *DEFA5*, *DEFA6*, *REG3G*, *REG3A*, and *PRSS2* and co-expressed or LI-specific AMPs *LYPD8*, *WFDC2*, *PI3* in different organs, with much lower chromatin accessibility of AMP genes in liver cells than that in ileal and colon cells (Fig. 5e), indicating that chromatin accessibility limits the regulation of intestinal AMPs by BATFs within the liver at the pre-transcriptional level. An equal number of cells from each of the three organs was randomly sampled ($n = 2500$ cells, respectively) to avoid any bias due to differences in cell counts between organs. After random sampling, the results were consistent with those obtained using all cells (Supplementary Fig. 9b, c). In addition, taking *DEFA5* and *WFDC2* as examples, the chromatin accessibility of the SI-specific AMP gene promoter region was high, and LI-specific AMP accessibility was low in ileal epithelial cells, while the opposite was true for colonic epithelial cells (Fig. 5f–i). Together, these results suggest that chromatin accessibility determines the potential of BATFs to control AMPs at the pre-transcriptional level, thus shaping the regional heterogeneity of intestinal AMPs (Fig. 5j).

Furthermore, the single-cell chromatin accessibility map also reasoned why ileal enterocytes express *NR1H4* at similar levels to Paneth cells but relatively low levels of SI-specific AMPs (Fig. 3f): Chromatin accessibility of SI-specific AMPs also varies between different epithelial cell types within the same intestinal region, and the chromatin accessibility of SI-specific AMPs in *APOA1* + *APOA4*+ enterocytes is much lower than that in Paneth cells (Supplementary Fig. 9d, e).

## Establishment of antimicrobial barrier during fetal life is dependent on BATFs-AMPs axis

How the fetal gut establishes antimicrobial immunity against inflammatory damage caused by the first colonization of the flora at delivery within a prenatal environment lacking microorganisms is unclear[17–19]. We hypothesized that the BATFs-AMPs axis is involved in forming AMP barrier during fetal development. We thus used a human intestinal epithelial cell developmental atlas including embryonic stem cell-derived human intestinal organoids (HIOs), human developing fetuses at 6–22 weeks post-conception (PCWs) and postnatal healthy human epithelial cells (Fig. 6a, b) to investigate the relationship between BATFs and the formation of AMP barrier during development.

We observed that the expression of both SI-specific and LI-specific AMPs in the fetal intestine and HIOs with a low level of BATFs was much lower than that in the postnatal period, especially *NR1H4* and *VDR* (Fig. 6c and Supplementary Fig. 10a–c). Moreover, the number of AMP-expressing epithelial cells was also much lower than that in the postnatal period (Fig. 6d). Regional heterogeneity of AMPs exhibited by the postnatal stage has not yet been established in early fetal development (6–8 PCW) (Fig. 6e, f and Supplementary Fig. 10d). Since 9 PCW, the expression levels of the SI-specific AMPs *DEFA5*, *DEFA6*, and *REG3G* and the LI-specific AMP *WFDC2* gradually increased in the ileum and colon, respectively, along with the increased expression levels of BATFs, *NR1H4*, *NR1H3*, and *VDR* (Fig. 6e–g and Supplementary Fig. 10d–f) while not Wnt signaling, another pathway that may regulate AMPs in a sterile environment (Fig. 6h). More importantly, we observed that the proportion of *DEFA5*+ non-Paneth cells but not Paneth cells increased

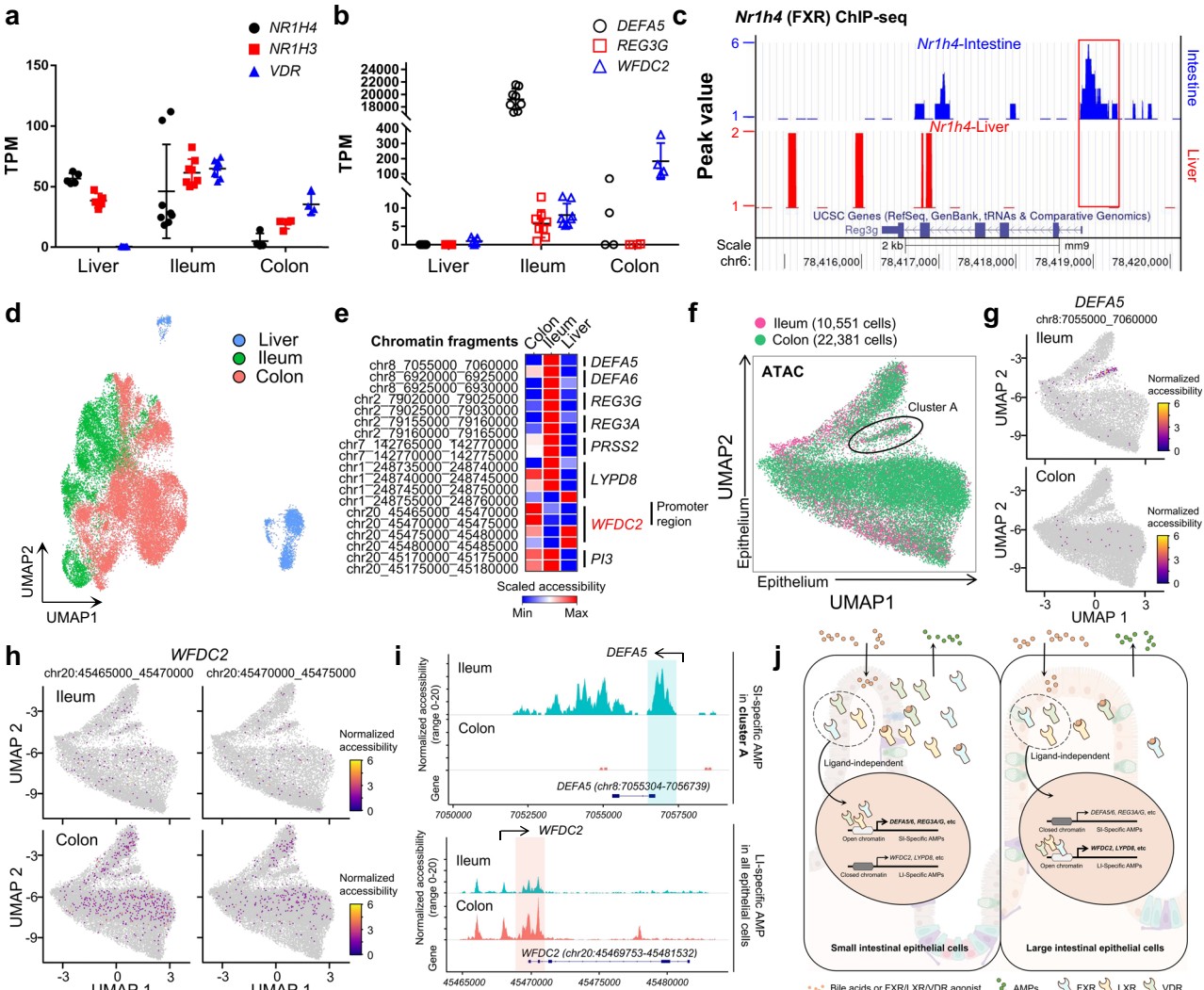

**Fig. 5 | Chromatin accessibility is involved in shaping the regional heterogeneity of AMPs. a, b** Shown are the expression of genes encoding BATFs (A) and AMPs (B) in human liver ($n = 5$ biologically independent samples), ileum ($n = 8$ biologically independent samples), and colon tissue ($n = 4$ biologically independent samples). **c** Intestinal (top) and hepatic (bottom) ChIP-seq analysis of FXR binding sites in *Reg3g*. **d** UMAP embedding of epithelial cells from liver (*ALB +*) and intestine (*EPCAM*+) colored by organ (liver, blue; ileum, green; colon, red). **e** Accessibility of AMP-associated gene fragments in different organs. The mean accessibility per organ is indicated with a color scale from 0 (closed) to 1 (open).

**f** UMAP embedding of epithelial cell profiles colored by organ (ileum, red; colon, green). **g, h** Accessible chromatin at AMP-related loci, assayed by scATAC-seq. Accessibility in ileum (top) or colon (bottom) epithelial cells of *DEFA5* (**g**) and *WFDC2* (**h**) fragments. **i** Aggregated single-cell profiles showed significant SI- (i.e. *DEFA5*) (top) and LI-specific AMP (i.e. *WFDC2*) (bottom) genes accessibility differences in different intestinal regions. **j** Schematic of the mechanism of BATFs and chromatin accessibility regulating regional heterogeneity of AMPs. Created with BioRender.com. All data are mean ± SEM. Source data underlying Fig. 5a, b are provided as a Source Data file.

during development (Supplementary Fig. 10g), which further eliminated the effect of cell differentiation on the expression of SI-specific AMPs. Overall, these results indicate that the BATFs-AMPs axis may be involved in establishing prenatal intestinal AMP barrier.

**Abnormal BATFs-AMPs axis mediates disruption of AMPs in CD**

Disturbances in Paneth cells and AMPs have been observed during CD[4,42]. We used scRNA-seq to investigate the association of BATFs with abnormal expression of AMPs in the ileal mucosa during CD (3 disease states and 40,434 cells in total) (Fig. 6i). We observed that the down-regulation of *NR1H4, NR1H3, and VDR* was accompanied by significant decreases in the expression of SI-specific AMPs (*DEFA5, DEFA6, PRSS2, REG3G,* and *ITLN2*) as CD progressed (Fig. 6j–l and Supplementary Fig. 11a−c), indicating that the disordered BATFs-AMPs axis may be involved in the collapse of intestinal antibacterial ability during CD.

At a single-cell level, we further observed that the proportions of *DEFA5*+ non-Paneth cells and specialized Paneth cells in the

inflammatory region of the CD ileum were significantly lower than those in healthy humans (Fig. 6m). During CD, consistent with the SI-specific AMPs, the BATFs were down-regulated in all epithelial lineages of the ileum, especially in stem cells that expressed high level AMPs in healthy state (Fig. 6n, top and Supplementary Fig. 11d). The BATFs of Paneth cells showed a downward trend, but no significance except *VDR* (Fig. 6n, bottom and Supplementary Fig. 11e). In addition, we also investigated the changes in AMP expression in immune cells during CD. Taking T / ILCs compartment as an example, consistent with epithelial cells, the down-regulation of *NR1H4* and *NR1H3* in ileal immune cells was accompanied by the decrease of SI-specific AMPs with the progression of CD (Supplementary Fig. 11f, g). The above results highlight a potential causal relationship between disrupted BATFs-AMPs axis and reduced SI-specific AMPs, and indicate that disturbance of BATFs may have compromised the anti-bacterial capacity of multiple cell lineages within the ileum, not just Paneth cells.

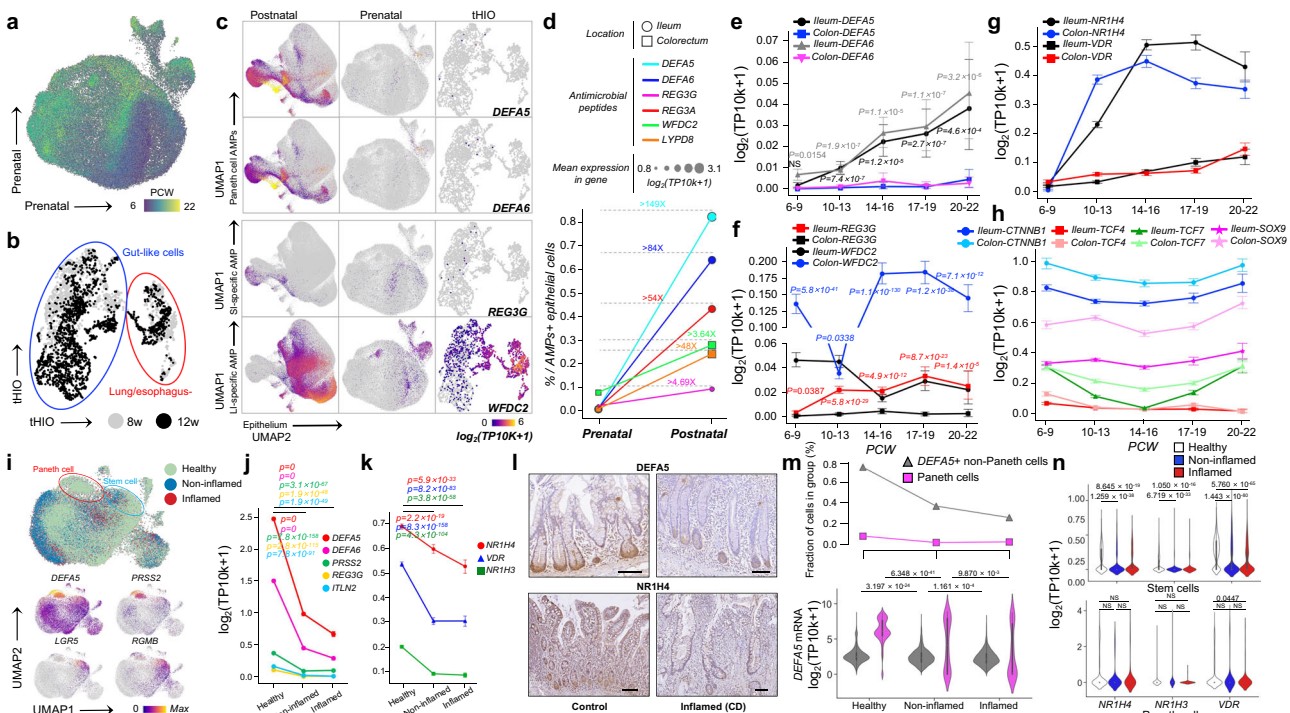

**Fig. 6 | BATFs-AMPs axis mediates the establishment of intestinal AMPs immune surveillance before birth and the disorder during CD. a, b** UMAP embedding of prenatal (**a**) and tHIO (**b**) scRNA-seq data colored by age. **c** Feature plots showing expression of selected AMP genes are different between postnatal, prenatal and tHIO. **d** Proportion of cells expressing AMP genes and expression level at pre- and postnatal stage. **e–h** Line charts showing the changes of the expression of AMPs (**e, f**), BATFs (**g**), and major genes of WNT signaling (**h**) in ileum and colon with fetal development (n = 7997 and 3386 cells for ileum and colon in 6–9 PCW respectively, n = 11,856 and 9147 cells for ileum and colon in 10–13 PCW respectively, n = 8382 and 5510 cells for ileum and colon in 14–16 PCW respectively, n = 2746 and 4856 cells for ileum and colon in 17–19 PCW respectively, n = 935 and 1835 cells for ileum and colon in 20–22 PCW respectively). **i** UMAP embedding of ileal epithelial cells colored by disease state (Healthy, green; Non-inflamed, blue; Inflamed, red) (top), and marker genes of stem cell (i.e. *LGR5* and *RGMB*) and Paneth cell (i.e. *DEFA5* and *PRSS2*) (bottom). **j, k** SI-specific AMPs (**j**) and BATFs (**k**) in ileal epithelial cells decreased significantly with the progression of CD (n = 24,094,

11,573, and 4767 cells for healthy, non-inflamed, and inflamed stats respectively). **l** Expression of DEFA5 and NR1H4 in histological sections (n = 9 for each phenotype). Scale bars, 100 μm. **m** Shown are alterations in the numbers of Paneth cells and other epithelial cells that expressing *DEFA5* (top) and mRNA level of *DEFA5* (bottom) during CD progression (n = 17,920 cells for *DEFA5*+ non-Paneth cells in healthy state, n = 4281 cells for *DEFA5*+ non-Paneth cells in non-inflamed state, n = 1235 cells for *DEFA5*+ non-Paneth cells in inflamed state, n = 1,958 cells for Paneth cells in healthy state, n = 222 cells for Paneth cells in non-inflamed state, n = 122 cells for Paneth cells in inflamed state). **n** Shown is the expression profile of BATFs in stem cells and Paneth cells during CD progression (n = 3244, 1435, and 817 cells for stem cells in healthy, non-inflamed, and inflamed states respectively, n = 1958, 222, and 122 cells for Paneth cells in healthy, non-inflamed, and inflamed states respectively). The length of the error bars is a 95% confidence interval for the mean in Fig. 6. NS, not significant. All p-values were calculated and reported using two-tailed Student's t-test.

## Discussion

Considering the great diversity of microbial communities in different anatomical regions of the gut[26], host antimicrobial barriers may be also regional heterogeneous. Here, we clarified the regional heterogeneity of intestinal AMPs at the single-cell level, and elucidated how regional AMP patterns are shaped at the transcriptional and pre-transcriptional levels.

Our data revealed an evolutionarily conserved AMP regional heterogeneity across species. Three regional AMP patterns in the human gut were uncovered, which included SI-specific AMPs, LI-specific AMPs, and co-expressed AMPs in both SI and LI. *LYZ, DEFA5, DEFA6, REG3A, PRSS2, ITLN2* and other SI-specific AMPs have been considered specific markers of Paneth cells (especially *LYZ* and *DEFA5/6*) and have been used as indicators of Paneth cell in numerous studies[43,44]. However, in this study, we unexpectedly found that α-defensin (*DEFA5/6*), with the highest expression in Paneth cells, was also highly expressed in other small intestinal epithelial cells (such as *LGR5*+ stem cells), even widely expressed in both epithelial and immune cells in the distal SI. Surprisingly, *LYZ* was strongly expressed in the epithelial cells of the proximal SI, resulting in a higher level of *LYZ* than in the ileum even with fewer Paneth cells. In addition to revealing the regional heterogeneity of AMPs, the significance of these

findings, which build on the multi-omics data (scRNA-seq, bulk RNA-Seq, RNAScope and Immunofluorescence), is that: Firstly, our findings raise concerns about the conclusions of previous studies on the differentiation and function of Paneth cells. For example, when *Defa5* is used as the specific marker to knockout the target gene in Paneth cells, it will certainly lead to gene knockout in *Defa5*-expressing *Lgr5*+ stem cells. Secondly, we found more specific markers for Paneth cell, such as *ITLN2* and *PRSS2* in human and *Defa3, Mmp7* in mouse, which should be considered to avoid off-target effects in gene knockout mice. Finally, cross-species study should be conducted in the future as did in recent research on intestinal stem cells[45], to confirm if these results, which are mostly based human data, also apply to mouse models.

The molecular mechanisms underlying regional heterogeneity of AMPs remains largely unclear. Our in silico analysis of TFs-target genes network[27], ChIP-seq of BATFs, and transcriptomic data from transgenic mice indicated a direct regulation of AMPs by BATFs, *NR1H4, NR1H3*, and *VDR*. Unexpectedly, our data suggested that the pharmacological agonists of three BATFs did not up-regulated the expression of AMPs, suggesting that BATFs regulate AMPs in a ligand (i.e., BAs)-independent manner. The ligand-independent regulatory capacity of BATFs has been well documented[31–33]. However, previous studies seem to show an important role played by BAs in the regulation of AMPs.

CDCA, CA, an endogenous agonist of FXR, upregulated α-defensins in murine ileal explants[13]. Another study reported seemingly contradictory results, with DCA inhibiting ileal Paneth cell function through activating the FXR signaling[36]. Our results clarified the effect of BAs on AMPs that CDCA and CA did not cause significant changes in α-defensins. DCA did not cause Paneth cell defects through FXR signaling, but inhibited the differentiation of secretory epithelial cells by perturbing WNT signaling, thus leading to the down-regulation of Paneth cell-AMPs. Collectively, these data suggest that BATFs regulate AMPs in a ligand-independent manner.

Nevertheless, the proposed BATFs-AMPs axis that is independent of BAs, still cannot answer the following three questions: why there are entirely different AMP patterns in SI and LI, why liver tissues with high expression of BATFs did not express intestinal AMPs, and why ileal enterocytes express NR1H4 at similar levels to Paneth cells but relatively low levels of SI-specific AMPs. By employing ChIP-seq and scATAC-seq data, we found that the existence of pre-transcriptional regulatory that limited the potential of BATFs in regulating different AMPs in different organs, and further revealed that chromatin accessibility controls the types of AMP coding genes that can be bound and up-regulated by BATFs in different intestinal regions and liver. In addition, the chromatin accessibility of SI-specific AMPs in ileal enterocytes is much lower than that in Paneth cells, suggesting that chromatin accessibility is also involved in the cell-type specificity of AMPs. Collectively, we revealed that the BATFs regulate AMPs in a ligand-independent manner and shape the regional heterogeneity of AMPs under the pre-transcriptional regulation of chromatin accessibility. The mechanisms underlying the differences in the chromatin accessibility of AMP genes in different regions have not been clarified in this work. A small number of cis-regulators known as pioneer TFs are thought to bind directly to chromosomes and actively alter chromatin structure in an environment where chromatin space and biological structures are in shutdown[46]. Future studies of pioneer TFs of AMP genes may uncover new regulators, further elucidate the mechanisms shaping the regional heterogeneity of AMPs.

Given the obvious correlation between regional heterogeneity of AMPs and microbial communities[26], there is a potential causal relationship between the spatial distributions of AMPs and microorganisms. The fetal data demonstrated that the regional AMP immunosurveillance is established within the sterile prenatal gut[17–19], implying that AMP regional heterogeneity is responsible for, rather than a consequence of, the regional heterogeneity of microbial community in the gut. In the future work, genetic perturbation models of AMP genes could be used to confirm the causal relationship between region-specific AMPs and microbial composition[43].

Considering the consistent expression of BATFs and SI-specific AMPs during the fetal development and the progression of CD, the proposed BATFs-AMPs axis may be involved in the establishment of antimicrobial barriers of fetal gut and the defects in antibacterial ability of multiple lineages during CD, which needs to be further confirmed by models of epithelial-specific knockout of BATFs in the future. In addition, the stability of FXR is largely regulated by acetylation, while SIRT1 is a key regulator of FXR deacetylation[47], suggesting that inhibition of SIRT1 will increase the stability of FXR protein. Recent studies have shown that intestinal epithelial cell-specific knockout of *Sirt1* leads to a significant upregulation of multiple SI-specific AMPs in the SI[48]. Collectively, SIRT1 may regulate the protein stability of FXR in epithelial cells through deacetylation to control the expression of intestinal AMPs. Therefore, SIRT1 inhibitors may become a promising approach to treat CD by restoring the homeostasis of AMPs and microbial communities.

Overall, our work answered two fundamental scientific questions, whether there is regional heterogeneity in the intestinal antimicrobial barrier and how it is regulated at the transcriptional and epigenetic levels, and uncovered several unexpected SI-specific AMP-expressing cells. These discoveries illuminate the logic underlying the regional heterogeneity of intestinal microbial community and the potential of BATFs in regulating distinct AMPs, providing rationale for restoring the flora homeostasis in different intestinal regions.

## Methods

### Single-cell RNA-seq and ATAC-seq datasets used in this study
We integrated our newly generated scRNA-seq data (available for download at https://data.mendeley.com/datasets/2d6z932mzw/1.) and 9 previously published sc(sn)RNA-seq and sci-ATAC-seq data to establish a landscape of intestinal epithelial cells across species, anatomical regions, developmental time, and health and disease states. These published data include: (1) HIOs derived from embryonic stem cells were subjected to single-cell transcriptome sequencing 4 and 8 weeks after in vivo transplantation into the kidney capsule of an immunocompromised mouse host (two samples in total; 10.17632/x53tts3zfr.5)[24]; (2) Fetal development data of proximal intestine, ileum, and large intestine ranged from 6 to 22 PCW (77 samples in total; GEO: GSE158702, https://doi.org/10.17632/x53tts3zfr.5, and https://www.gutcellatlas.org/)[15,16,24]; (3) Postnatal healthy human duodenum, jejunum, ileum, appendix, colon, and rectum scRNA-seq data (154 samples in total; GEO: GSE125970, Single Cell Portal: SCP259, https://doi.org/10.17632/x53tts3zfr.5, and https://www.gutcellatlas.org/)[15,22–24]; (4) Non-inflammatory and inflammatory mucosal biopsies data of ileal Crohn's disease (29 samples in total; https://www.gutcellatlas.org/, GEO: GSE134809)[15,21] and colorectal ulcerative colitis (84 samples in total; Single Cell Portal: SCP259)[22]; (5) SnRNA-seq data of mouse ileal and colonic epithelium (25 samples in total; Single Cell Portal: SCP1038)[20]; (6) ScATAC-seq data of postnatal human liver, ileum, and colon (11 samples in total; GEO: GSE184462)[49]. (7) ScRNA-seq data of 44 postnatal human tissues (GEO: GSE201333)[26]. It is worth noting that before integrating data from different sources, the gene symbol of all open-source human single-cell sequencing data is modified to the gene symbol of GRCh38.p13 human reference genome.

### Bulk RNA-seq and microarray data used in this study
The bulk RNA-seq data used in this study include (see also in Supplementary Data 4): (1) Bulk RNA-seq data of mice colon tissues in which *Nr1h4* was ablated in the intestine (*Nr1h4*-intKO: *Vill*-Cre; *Nr1h4*-floxed), the liver (*Nr1h4*-livKO: *Alb*-Cre; *Nr1h4*-floxed), or in full body (*Nr1h4*-totKO), and bulk RNA-seq data of *Nr1h4*-floxed mice colon tissues (*Nr1h4*-fl/fl) ($n = 3$ mice per genotype group; GEO: GSE163157)[29]; (2) Bulk RNA-seq data of male mouse ileum and colon tissues including *Vdr*-KO mice and transgenic mice with hVDR only expressed in the distal intestine (KO/*Vdr*-Tg). All the mice which are C57BL6J background will be raised until 2–3 months old with standard chow diet ($n = 3$ mice per genotype group; GEO: GSE144978)[30]; (3) Bulk RNA-seq data of healthy human liver, ileum, and colon tissue (five liver samples, four duodenum samples, eight ileum samples, and four colon samples; AYEXPRESS: E-MTAB-1733)[25]; (4) Microarray data of WT mice on chow and WT mice on chow supplemented with FXR agonist PX20606 (PX) compound 10 mg/kg/day for 2 weeks. Small intestine (proximal and distal) were collected from all 8 mice ($n = 4$ mice per group; GEO: GSE74101)[34]; (5) Bulk RNA-seq data of mice ileal tissues in which synthetic LXR ligand GW3965 was suspended in 0.5% carboxymethyl cellulose and was orally administered twice weekly at 1 mg/kg/day for the last 5 weeks in the 10-week period following intestinal resection to the control ($n = 3$ mice per group; ENA: PRJNA705703)[35]; (6) Bulk RNA-seq data of male mouse ileal and colon tissues including control group without any injection and experimental group administered 1 ng/g bw 1,25(OH)2D3 at 48, 24 and 6 h prior to termination to determine both early and late effects of 1,25(OH)2D3 ($n = 3$ mice per group; ENA: PRJNA605550)[30]; (7) Primary mouse ileal organoids were treated with vehicle or a designer bile acid pool for 24 h. 4 distinct bile acid pools were used, which varied based on the species we modeled (human or

mouse) and the proportion of 12a-hydroxylated bile acids (10% or 90%). Three samples from each treatment group were submitted for RNA sequencing (GEO: GSE144398)[50].

## ChIP-seq data used in this study

The ChIP-seq data used in this study include: (1) Genome-wide FXR binding in liver and small intestine of mice treated with a synthetic FXR ligand (GW4064) by ChIP-seq (available at http://genome.ucsc.edu/goldenPath/customTracks/custTracks.html#Mouse)[51]; (2) Genome-wide VDR binding in proximal small intestine of mice treated with VDR ligand (1,25(OH)2D3, 10 ng/g bw) or vehicle control by ChIP-seq (available for download at GEO: GSE69179)[52]; (3) Genome-wide LXR binding in HT29 cells (Colorectal Adenocarcinoma) treated with vehicle control (DMSO) or after drug treatment (GW3965 and rosiglitazone) by ChIP-seq (available for download at GEO: GSE77039)[53].

## Human specimens for single-cell experiment

We generated scRNA-seq profiles from 14 intestinal samples that were collected from 10 healthy donors recruited in Qilu hospital of Shandong University at the time of routine gastroscopy or enteroscopy or colonoscopy (see also in Supplementary Data 1). Healthy volunteers were individuals without gastrointestinal tumors, polyps or other organic diseases, and who were overall healthy with no underlying diseases such as hypertension and diabetes. All samples were obtained with informed consent, and the study was approved by the medical science research ethics committee of Qilu hospital of Shandong University (KYLL-202008-127-1). All relevant ethical regulations of the medical science research ethics committee of Qilu hospital were followed.

## Animal experiments

Animal experiments were carried out in compliance and approved by the Shandong university Specific Pathogen Free (SPF)-animal Center. All experimental animal procedures were approved by the Animal Care and Animal Experiments Committee of Shandong university (ECSBMSSDU2020-2-057). Six to eight weeks old male C57BL/6 mice were obtained from Nanjing GemPharmatech animal center and maintained in SPF facilities at Shandong University. Isolation of mouse intestinal lamina propria cells was done as previously described.

## Single-cell collection and sorting

Intestinal mucosa was freshly sampled from the duodenum, jejunum, ileum of the volunteers, and intestinal biopsies were washed in PBS to remove mucus and blood cells. Intestine samples were then incubated with shaking in PBS containing 10 mM EDTA and 20 mM HEPES at 37 °C for 20 min. After shaking, crypts and villus fraction in the medium were mechanically detached, strained, washed, and centrifuged; the pellet was then resuspended in warm TrypLE Express (GIBCO) and digested to single epithelial cells at 37 °C. To obtain the lamina propria cell compartment, the rest pieces were digested by shaking in 2 mL of 5% (v/v) FBS in RPMI medium containing DNase I (Sigma) (150 μg/ml) and collagenase IV (Sigma) (1.5 K U/ml) at 37 °C for 20–30 min. The digested tissue was homogenized by vigorous shaking and filtered through 100 μm cell strainer. After centrifugation, the pellets were harvested and resuspended in a complete cell medium. Single-cell suspensions of epithelial and lamina propria cell components were pelleted, washed, strained, and resuspended in FACS buffer. 7-aminoactinomycin D (7-AAD) was added just before flow sorting. 7-AAD-negative living cells of epithelial and lamina propria compartment were sorted for further single-cell mRNA-sequencing separately. Data for all sorted cells were recorded for later experiments.

## Library preparation and single-cell RNA sequencing for 10X Genomics single cell platform

Cells were concentrated to 700–1000 cells/μL and loaded on Gem-Code Single Cell Instrument (10x Genomics; Pleasanton, CA, USA) to generate single-cell gel bead-in-emulsions (GEMs). Next, GEMs were subjected to library construction using Chromium Single Cell 3' Reagent Kits v2 (10x Genomics; Pleasanton, CA, USA) according to the manufacturer's instructions, the steps of which included incubation at room temperature, complementary DNA amplification, fragmentation, end repair, A-tailing, adapter ligation, and sample index polymerase chain reaction. To be compatible with BGISEQ-500 sequencing platform, libraries conversion was performed using the MGIEasy Universal Library Conversion Kit (App-A) (Lot: 1000004155, BGI). Then the converted library was subjected to subsequent DNA circularization and rolling-cycle amplification to generate DNA nanoballs. Purified DNA nanoballs were sequenced using the BGISEQ-500 sequencing platform, generating reads containing 16 base pairs of 10xTM barcodes, 10 base pairs of UMIs, and 100 base pairs of 3' complementary DNA sequences.

## Library preparation and single-cell RNA sequencing for DNBelab C single cell platform

DNBelab C Series High-throughput Single-cell System (BGI-research) was utilized for scRNA-seq library preparation. In brief, the single-cell suspensions were converted to barcoded scRNA-seq libraries through steps including droplet encapsulation, emulsion breakage, mRNA captured beads collection, reverse transcription, cDNA amplification and purification. cDNA production was sheared to short fragments with 250–400 bp, and indexed sequencing libraries were constructed according to the manufacturer's protocol. Qualification was performed using Qubit ssDNA Assay Kit (Thermo Fisher Scientific) and Agilent Bioanalyzer 2100. All libraries were further sequenced by the DIPSEQ T1 sequencing platform (China National GeneBank) with pairend sequencing. The sequencing reads contained 30-bp read 1 (including the 10-bp cell barcode 1, 10-bp cell barcode 2 and 10-bp unique molecular identifiers (UMI)), 100-bp read 2 for gene sequences and 10-bp barcodes read for sample index.

## RNAscope in situ hybridization

Human intestinal samples were fixed in 4% formaldehyde solution, dehydrated with ethanol and embedded in paraffin. Four um-thick tissue sections on glass slides and the following staining was performed with the RNAscope manual assay using the Multiplex Fluorescent Detection Kit v2 (Advanced Cell Diagnostics, Cat#323110) according to manufacturer's protocols. Briefly, deparaffinized slides were pretreated with hydrogen peroxide (Advanced Cell Diagnostics, Cat#322381), followed by permeabilization in target retrieval reagent (Advanced Cell Diagnostics, Cat#322000) for 15 min, and digestion with Protease Plus (Advanced Cell Diagnostics, Cat#322381) at 40 °C for 30 min. A combination of probes for DEFA5 (Advanced Cell Diagnostics, Cat#423981) and LGR5 (Advanced Cell Diagnostics, Cat#311021) detection was hybridized at 40 °C for 2 h. Signal amplification was followed by development of appropriate HRP channels with dyes Opal 520 (Asbio Tecchnology, Cat#ASOP520), Opal 690 (Asbio Tecchnology, Cat#ASOP690), and DAPI (Advanced Cell Diagnostics, Cat#323108) served as nuclear stain. Slides were mounted in Prolong Gold antifade reagent (Invitrogen, Cat#P36930). Confocal images were taken with the high speed confocal platform (Andor, Dragonfly 200).

## Immunofluorescence

Human intestinal samples were fixed in 4% formaldehyde solution, dehydrated with ethanol, and embedded in paraffin. Four um-thick tissue sections on glass slides were deparaffinized through an ethanol gradient, and then tissue sections were incubated in retrieval solution for antigen retrieval at 95 °C for 20 min. The sections were permeabilized and blocked for non-specific binding with 5% BSA and 0.1% Triton X-100 in PBS for 1 h at room temperature. Then, the sections were incubated for 14 h with the primary antibody at 4 °C. The fluorescein-

labeled secondary antibodies for immunofluorescence were added for 1 h at 37 °C. Then, slides were mounted with Slowfade Mountant+DAPI (Life Technologies, S36964) and sealed.

## Immunohistochemistry

Slides preparation was described in Immunofluorescence part. Four um-thick tissue sections on glass slides were deparaffinized through an ethanol gradient, and then tissue sections were incubated in retrieval solution for antigen retrieval at 95 °C for 20 min, and then endogenous peroxidase was deactivated by incubating with 3% hydrogen peroxide for 10 min. The sections were permeabilized with 0.1% Triton X-100 in TBS for 10 min and blocked for non-specific binding with 5% BSA for 30 min at room temperature. Then, the sections were incubated for 14 h with the primary antibody at 4 °C. Sections were incubated with horseradish peroxidase-conjugated anti-rabbit antibody and stained.

## Organoid cultures and treatments

The distal small intestine of male mouse was isolated and dissected lengthwise, then the bowel segment was cut into 5 mm pieces. After washing in PBS, bowel fragments were incubated in chelation buffer (containing 2 mmol/L EDTA and 10 mmol/L HEPES) and shaken on a rocking platform for 30 min (4 °C, 100 rpm). Then, bowel pieces were resuspended in clean PBS and pipetted several times to isolate intestinal crypts. Isolated crypts were embedded in Matrigel Matrix (Corning) on ice and carefully seeded in a pre-warmed 48-well plate (Corning). After Matrigel solidified, IntestiCult™ Organoid Growth Medium (STEMCELL Technologies) supplemented with penicillin-streptomycin (100 units/100 μg/mL) was added and changed every 3–4 days.

Organoid treatment: For bulk RNA sequencing, organoids cultured for 2 days were incubated with DMSO and a series of bile acids at 50 mmol/L for 96 h and lysed for RNA extraction. Each treatment contained three replicate wells. For single-cell RNA sequencing, organoids cultured for 2 days were incubated with DMSO and DCA at 50 mmol/L for 96 h, and these organoids were then resuspended in warm TrypLE Express (GIBCO) and digested to single cells at 37 °C.

## Dataset integration

This work incorporates numerous open-source scRNA-seq datasets, which may confound biological effects when analyzing inter phenotypic differences due to batch effects in sample processing, sequencing pipelines between different sources. To alleviate the impact of batch effects, we referred to the batch effect solution of a high-quality single-cell meta-analysis study, which integrated the macrophages from 14 organs of 41 public datasets[54]. Specifically, we adopted three steps that introduced additional limitations while mitigating batch effects. Firstly, we only integrated datasets that were open source in the form of raw counts by using 10x Genomics single-cell sequencing technology, excluded datasets that were subjected to normalization of expression matrix, and datasets with <20,000 genes. Secondly, when merging the gene expression matrices of different datasets, only the genes included within all datasets were taken forward. Thirdly, the total counts were normalized to 10,000 reads per cell, so that counts become comparable among cells from different datasets.

## Alignment, quantification, and quality control of single-cell RNA sequencing data

Droplet-based sequencing data were aligned and quantified using the CellRanger software (version 3.0.2 for 3' chemistry) using the GRCh38.p13 human reference genome. Scanpy (version 1.7.1) python package[55] was used to load the cell-gene count matrix and perform quality control for newly generated dataset and collected datasets. For each sample, after removing the mitochondrial (gene symbols start with MT-) and ribosomal Protein (gene symbols start with RP) genes, cells with fewer than 2000 UMI counts and 250 detected genes were considered as empty droplets and removed from the datasets. After that, genes expressed in fewer than three cells were discarded.

## Doublet detection

To exclude doublets, we applied Scrublet software (version 0.2.3)[56] to identify artifactual libraries from two or more cells in each scRNA-seq sample, including newly generated dataset and collected datasets. The doublet score for each single cell and the threshold based on the bimodal distribution was calculated with default parameters (sim_doublet_ratio=2.0; n_neighbors=None; expected_doublet_rate=0.1, stdev_doublet_rate=0.02). All remaining cells and cell clusters were further examined to detect potential false-negatives from scrublet analysis according to the following criteria: (1) Cells with >8000 detected genes, (2) Clusters that expressed marker genes from two distinct cell types, which are unlikely according to prior knowledge (i.e. CD3D for T cells and EPCAM for Epithelial cells). All cells or clusters flagged as doublets were removed from further downstream analysis.

## Graph clustering and partitioning cells into distinct compartments

Downstream analysis included normalization (scanpy.pp.normalize_total method, target_sum=1e4), log-transformation (scanpy.pp.log1p method, default parameters), cell cycle score (scanpy.tl.score_genes_cell_cycle method), cell cycle genes defined in Tirosh et al, 2016[57], feature regress out (scanpy.pp.regress_out method, UMI counts, percentage of mitochondrial genes and cell cycle score were considered to be the source of unwanted variability and were regressed), feature scaling (scanpy.pp.scale method, max_value = 10, zero_center=False), PCA analysis (scanpy.tl.pca method, svd_solver='arpack'), batch-balanced neighborhood graph building (scanpy.external.pp.bbknn method, n_pcs=20)[58], leiden graph-based clustering (scanpy.tl.leiden method, Resolution=1.0)[59], and UMAP visualization[60] (scanpy.tl.umap method) performed using scanpy. Clusters were preliminarily partitioned into 6 compartments, using marker genes found in the literature in combination with differentially expressed genes (scanpy.tl.rank_gene_groups method, method='Wilcoxon test'). Specifically, epithelial compartment was annotated using a gene list (EPCAM, KRT8, KRT18, KRT19, PIGR), T and ILCs compartment (CD2, CD3D, CD3E, CD3G, TRAC, IL7R), B cell compartment (JCHAIN, CD79A, IGHA1, IGHA2, MZB1, SSR4), MNPs compartment (HLA-DRA, CST3, HLA-DPB1, CD74, HLA-DPA1, AIF1), Mast cell compartment (TPSAB1, CPA3, TPSB2, CD9, HPGDS, KIT), and Stromal cell compartment (IGFBP7, IFITM3, TCF7L1, COL1A2, COL3A1, GSN). Then, the epithelial compartment was sorted for downstream analysis.

## Define cell subsets in different epithelial compartments

Re-clustering and differential gene expression analysis (scanpy.tl.rank_genes_groups method, method='wilcoxon') were performed on each epithelial cell compartment to accurately identify cell types or subsets and characterize the differential genes of each subset (Supplementary Fig. 1a–c).

For the postnatal human epithelial cells, the results of preliminary re-clustering indicated that some lineages from different intestinal regions, such as absorptive enterocytes from proximal SI, distal SI, and LI were independent of each other in UMAP and cannot be grouped into a cluster. Therefore, we performed clustering and annotation on epithelial compartment of proximal SI, distal SI, and colorectum respectively, revealing 14 cell types and their proportional distribution and differential genes. For prenatal human epithelial cells, we first integrated fetal scRNA-seq data with postnatal data on defined cell types and performed dimensionality reduction, clustering, and visualization to enable label transfer of fetal cells with the help of postnatal epithelial cell identity definitions. Epithelial lineage cells shared between pre- and postnatal data were stem cells (LGR5, ASCL2, SMOC2, RGMB)[15,22,24], transit-amplifying cells (TA; MKI67, TOP2A)[15,22,24], Paneth

cells (*DEFA5, DEFA6, REG3A, PRSS2*)[15], goblet cells (*CLCA1, FCGBP, ZG16, MUC2*)[15,22,24], *BEST4*+ enterocytes (*BEST4, OTOP2*)[15,22], immature enterocytes (*OLFM4, DMBT1*)[15,22], *APOA1*+ enterocytes (*APOA1, ANPEP, FABP2*)[15], *CA1*+ colonocytes (*CA1, CA2, SLC26A3*)[15,22], *PGC*+ mucous cells (*PGC, TFF2, MUC6*)[61], enteroendocrine cells (*CHGA, CHGB, NEUROD1*)[15,22,24], microfold cells (*CCL20, CCL23, MIA*)[15,22,24], Tuft cells (*IRAG2, SH2D6, AZGP1*)[15,22,24], *PI3*+ enterocytes (*PI3, CD74, LCN2*), *IGKC*+ enterocytes (*IGHA1, IGHA2, JCHAIN, IGKC*) (Fig. 1c and Supplementary Fig. 1a). It should be noted that we also identified a small number of cells expressing Paneth cell markers in colon and rectum tissues derived from healthy individuals (7 cells in the colon, 1 cell in the rectum). Given that the right colon does normally have a small number of Paneth cells[62,63], as well as the expression all Paneth cell markers (*CCL24, DEFA5, DEFA6, REG3A*) in these cells, they were retained and annotated as Paneth cells. The single cell in the rectum only expressed *DEFA6* was removed.

For the human intestinal organoid compartment, the cells were preliminarily divided into gut-like cells and lung/esophagus-like cells based on the marker genes. Then, the gut-like cells were clustered into 7 clusters and annotated as stem cells[15,22,24], immature enterocytes[15,22], enterocytes[15], *BEST4*+ enterocytes[15,22], *PGC*+ mucous cells[61], enteroendocrine cells[15,22,24], and goblet cells[15,22,24]the lung/esophagus-like cells were clustered into 2 clusters, basal like cells (*KRT4, TP63*)[24] and ciliated like cells (*FOXJ1, PIFO*)[24] (Fig. 1c and Supplementary Fig. 1b).

For the mouse intestinal epithelial cells, we respectively performed clustering and annotation on epithelial compartment of ileum and colon tissue, revealing 12 cell types and their proportional distribution and differential genes[20,64] (Fig. 1d and Supplementary Fig. 1c).

## Transcription factor module analysis

Python package pySCENIC workflow (version 0.11.0) with default settings was used to infer active TFs and their target genes in all human cells[27]. In brief, the pipeline was implemented in three steps. First, a single-cell gene expression matrix was filtered to exclude all genes detected in fewer than ten total cells, and the remaining genes were used to compute a gene-gene correlation matrix for co-expression module detection using a regression per-target approach GRNBoost2 algorithm. Second, we pruned each module based on a regulatory motif near a transcription start site (TSS). Cis-regulatory footprints could be obtained with positional sequencing methods (for example, from ChIP-seq motif calling with an antibody against a TF). Binding motifs of TFs across multiple species were then used to build an RCisTarget database[65]. Precisely, modules were retained if the TF-binding motif was enriched among its targets, while target genes without direct TF-binding motifs were removed. Third, we scored the impact of each regulon for each single-cell transcriptome using AUC score as a metric by the AUCell algorithm.

Each step of this pipeline used rank statistics, and the last classification step ran independently for each cell, avoiding a batch effect. The transcription factor motif scores for gene promoters and around transcription start sites for hg38 human reference genome were downloaded from the RcisTarget database[65], and the TF gene list was downloaded from Humantfs database[66]. GRN plots of the SI-specific AMPs (Supplementary Fig. 7b) were done using the Evenn software[67].

## Fate decision tree construction (regulon-based)

Dendrogram plots were constructed for epithelial cells using Scanpy (sc.pl.dendrogram method) on the AUCell matrix of 608 regulons to observe subtler changes, respectively (Supplementary Fig. 7a). We deciphered the diverging composite rules of a regulon-based dendrogram by testing each branching node for differential regulon importance. Therefore, we performed differential regulon expression analysis of every node with Wilcoxon test (sc.tl.rank_gene_groups method, method='Wilcoxon test') to derive the action propagation program of the regulons.

## Trajectory analysis

We applied the pseudo-time analysis to infer the differentiation trajectories of absorptive enterocytes across different intestinal regions (Fig. 3b), and then identified the influence of the regional microenvironment on the differentiation and functional shifts of epithelial cells. The analysis pipeline was implemented in two steps. First, clustering and annotation were performed on epithelial cells of the proximal small intestine, distal small intestine (ileum), and colorectum, respectively. Second, we set *LGR5*+ stem cells as the initial point for differentiation and implemented pseudo-time analysis absorptive enterocytes (included immature enterocytes, *APOA1*+ enterocytes and *CA1*+enterocytes) with diffusion map (sc.tl.diffmap method, default parameters), Partition-based graph abstraction (PAGA) (sc.tl.paga method)[68] and Force-directed graph drawing algorithm (sc.tl.draw_graph method, init_pos='paga')[69].

## Differentiation dynamics of antimicrobial peptides expression

We implemented the Wishbone python package (scanpy.external.tl.wishbone method, default parameters)[70], an external module to the Scanpy package, to identify the dynamic trajectory of epithelial cell differentiation of proximal small intestinal and the expression of AMPs along the trajectory (Fig. 2e). In brief, we processed the normalized gene-cell matrix in three steps: (1) the principal component analysis and batch correction based on BBKNN[58] were performed; (2) we estimated the diffusion map of epithelium differentiation; (3) Wishbone[70] and Phonograph[71] python package were used to determine the differentiation branch and cluster the trends of AMPs.

## Scoring gene set and identifying significant changes

We scored gene sets of all cells and clusters using the Scanpy python package (sc.tl.score_genes method, ctrl_size=len(genesets), gene_pool=None, n_bins=25, use_raw=None). The score was the average expression of a set of genes subtracted from the average expression of a reference set of genes. The reference set was randomly sampled from the gene_pool for each binned expression value. To prevent highly expressed genes from dominating a gene set score, we scaled each gene of the log2 (TP10K + 1) expression matrix by its root mean squared expression across all cells. After obtaining the signatures score-cell matrix, differential signature analysis (sc.tl.rank_gene_groups method, method='Wilcoxon test') was implemented to identify significant changes among different intestinal regions. All pathways used in gene set enrichment analysis of regional absorptive enterocytes (Fig. 3c) were obtained from Reactome[72].

## ScATAC-seq data analysis

We downloaded fragment files for human liver (1 sample), ileum (3 samples), and colon (7 samples) sci-ATAC-seq datasets and combined the 11 files into a single fragment file, adding a prefix to the cell barcodes to mark which cell originated from which dataset. We called peaks using the combined dataset with MACS3 (version 3.0.0a7), using the CallPeaks function in Signac (R, version 1.5.0)[73]. Peaks overlapping genomic blacklist regions for hg38 were then removed, resulting in a set of 241,385 peak regions. Then, we quantified counts in peaks using the FeatureMatrix function in Signac and removed cells with (1) < 1000 nCount_peaks; (2) > 100,000 nCount_peaks; (3) < 2 nucleosome_signal; (4) > 1 TSS.enrichment. After that, we reduced the dimensionality by applying LSI and UMAP, using LSI components 2 to 30, with harmony for performing dataset integration. Finally, we generated 'pseudo-bulk' accessibility tracks grouped by cluster around specific genomic regions (i.e. the AMP genes) using the CoveragePlot function in Signac package (Fig. 5 and Supplementary Fig. 9b−e).

In addition, to obtain the chromatin accessibility of TF-target gene binding sites, we used NCBI gene database to obtain the gene sequence of the promoter region of the target gene (2k bp upstream and 100 bp downstream of the gene transcription starting point), and

then predicted the TF binding sites using JASPAR 2022 database. The chromatin accessibility of binding sites was demonstrated by using the CoveragePlot function of Signac.

## Reporting summary

Further information on research design is available in the Nature Portfolio Reporting Summary linked to this article.

## Data availability

All scRNA-seq and bulk RNA-seq data generated in this study have been deposited in Mendeley Data (https://data.mendeley.com/datasets/2d6z932mzw/1). Enriched ligand-dependent nuclear receptors and chemicals in a specific region depicted by IPA are provided in the Supplementary Data 3. The published scRNA-seq and scATAC-seq data used in this study are available in GEO: GSE158702, GSE125970, GSE134809, GSE184462, GSE201333; Mendeley Data: https://doi.org/10.17632/x53tts3zfr.5; Single Cell Portal: SCP259, SCP1038; Gut cell atlas [https://www.gutcellatlas.org/]. The published bulk RNA-seq data used in this study are available in GEO: GSE163157, GSE144978, GSE74101, GSE144398; AYEXPRESS: E-MTAB-1733, ENA: PRJNA705703, PRJNA605550. The published ChIP-seq data used in this study are available in http://genome.ucsc.edu/goldenPath/customTracks/custTracks.html#Mouse, GEO: GSE69179, GSE77039. Source data are provided with this paper.

## Code availability

All the codes related to the analysis are publicly available at https://github.com/YueWang1997/NC_AMPs.

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

## Acknowledgements

We thank Translational Medicine Core Facility of Shandong University for consultation and instrument availability that supported this work. The scientific calculations in this paper have been done on the HPC Cloud Platform of Shandong University. This work was financially supported by National Key Research and Development Program of China (2020YFA0804400), National Natural Science Foundation (NSF) of China (82071854) and Youth Interdiscipline Innovative Research Group of Shandong University (2020QNQT009) to S.L.; NSF of China (82070552 and 82270580), National Clinical Research Center for Digestive Diseases supporting technology project (2015BAI13B07) and Taishan Scholars Program of Shandong Province to Y.L.; NSF of China (81770538) and Taishan Scholars Program of Shandong Province to X.Z.; NSF of China (82070540) to Y.Y.

## Author contributions

Y.L., X.Z. and S.L. initiated, designed and supervised the project; Y.Y., X.W. and LX.L. carried out human tissue collection; M.Z. and B.F. performed human tissue processing and scRNA-seq experiments; M.Z. performed flow cytometry and immunofluorescence validation in the human and mice tissue and data interpretation; Y.C. and S.C. performed immunohistochemistry and organoid experiments; Y.W. analysed sc(sn) RNA-seq, scATAC-seq, bulk RNA-seq, and ChIP-seq data and generated figures; Y.W., M.Z., J.Z., X.M. and H.G. contributed to data visualization; Y.W., Y.Y., LX.L., M.Z., J.Z., H.G., B.F., X.W., X.M., Y.C., Y.X., S.C., L.L., H.C., R.Z., M.M., Z.L., R.J., M.L., X.Y., X.Z., S.L. and Y.L. contributed to interpretation of the results; Y.W. wrote the original draft; Y.W., Y.Y., LX.L., M.Z., M.L., X.Y., X.Z., S.L., and Y.L. conducted the review and editing of the manuscript.

## Competing interests

The authors declare no competing interests.
