## [Peer Review File · Nature Communications]

Bile acid-dependent transcription factors and chromatin accessibility determine regional heterogeneity of intestinal antimicrobial peptidesREVIEWER COMMENTS

Reviewer #1 (Remarks to the Author):

The intestine is an organ for the absorption of nutrients which exists extensively host-microbe interaction. The organ has evolved diverse mechanisms against microbial invaders, including antimicrobial peptides. However, the heterogeneity in cell composition or function among intestine segments is not well resolved. In this manuscript, Wang et al. provided an intestinal cell landscape by integrating scRNA-Seq datasets produced by themselves or published before and described regional heterogeneity and cell type specificity of intestinal AMPs. Then they predicted that the BATFs were the upstream regulators of AMPs and chromatin accessibility determines the potential of BATFs and shapes the regional heterogeneity of intestinal AMPs. Finally, the authors infer the BATFs-AMPs axis participated in the establishment of intestinal antimicrobial barriers in fetuses and that the defects of BATFs mediate disruption of AMPs during Crohn's disease.

Overall, the topic is interesting and the content and quality of the manuscript are satisfactory. However, some points could be confirmed carefully. For example, the integration from different sources should set more strict filter criteria. More evidence should be provided to support the conclusion that immune cells also expressed high level of AMPs.

It is very interesting and efficient that the authors integrated amounts of published omics datasets to support the conclusion of the manuscript. But the filter and analysis should be described in more details. Necessary quality controls are often missed in integration. Moreover, the showing in the manuscript is very confusing in this version, FPKM, TPM, mRNA counts, and values with log-transformed...

Before the manuscript could be accepted by the journal, some suggestions could be considered.

Major points:

1. The authors did not set strict filter criteria for the integration of intestine cells from a different source. The integration often faced the problem of over-integration or not enough, the negative and positive control should be set. For example, cells from the liver could be used as a negative control. And cells with the same identity should be integrated into the same cluster derived from different patients. These controls were not shown in the manuscript. In addition, the marker genes for cell identity annotation in Fig. S1 should be given the corresponding references or sufficient evidence for identity definition.

2. In lines 112-123, the authors announced that the immune cells still expressed high levels of SI-specific AMPs. They should give more evidence. In my opinion, three points could be considered. 1) Sequencing reads from immune gene and AMP genes should be counted in the same cells but not in defined clusters to exclude the possibility of mis-annotated cells. 2) The immune cells in other organs or tissues should be checked to confirm whether the co-expression is specific in the small intestine. 3) The co-immunofluorescent assay for immune and AMPs in the same cell.

3. The normalization and showing method for RNA-Seq are inconsistent. FPKM and histogram in Fig. 3i - 3k; TPM and histogram with points in Fig 4d.e; raw mRNA counts and line chart in Fig 4f; TPM with log-transformed and histogram with points in Fig 4g; TPM with log-transformed and points in Fig 4h. The same questions also existed in other figures. I suggested the authors use the consistent TPM or FPKM and histogram with points to show these results. And all the published RNA-seq data re-analyzed with the same parameters. All the statistical tests should be performed on new results.

4. The control also should be used for scATAC-seq. In fig 5d, cells from liver overlapped with cells from two other organs in the mass. The authors should explain the causes of these results. Moreover, in my view, the cell from the liver shown in fig. 5d are less than other organs. Whether this cell number influences the peak levels shown in Fig. S4d and Fig 5e.

5. The author also discussed the relationship of the regional heterogeneity of AMPs and microbiota. Maybe the authors could explain this problem by re-analyzing the metagenome data of the intestine. Of course, this suggestion is not necessary.

Minor points:

1. The colors for cell type in Fig. 1b(i) are not distinguishable.

2. The figure legend for Fig 2a should state three figures were produced separately.

Reviewer #2 (Remarks to the Author):

The manuscript presents an extensive characterization of human antimicrobial peptides (AMPs) expression in human and mouse gut. The authors performed trajectory inference based on single-cell transcriptomics to map the transcription factor regulatory landscape of the cell differentiation, and concluded that the bile acid-activated TFs are the upstream regulators of AMPs expression. The study further demonstrated that such regulation is ligand-independent, and highly depends on chromatin accessibility in a cell-type-specific manner. My concern relates to a few major claims that the authors made in the manuscript without providing conclusive evidences:

1) At Line 100 (Fig. 2c), the authors claimed that "LGR5+ stem cells and transit-amplifying (TA) cells in the human SI also expressed high levels of SI-specific AMPs (LYZ, DEFA5/6, REG3A/G, PLA2G2A, ITLN2, PRSS2)." But this is not upheld by the figures showed. In Fig. 2c, majority of the cell populations except Paneth cells from ileum has the minimal expression of these AMPs based on the scale bar. The stem cells and TA cells have the same low-level expression as all the enterocyte clusters based on the scale. Moreover, the staining pictures showed in Fig. 2g also indicated that the DEFA5-expressing cells are solely enriched in the crypt, not even in the TA zone. Most importantly, this manuscript included one of the recent dataset from "Cells of the human intestinal tract mapped across space and time (<https://doi.org/10.1038/s41586-021-03852-1>)", which clearly showed the specificity of these AMPs in Paneth cells in their original analysis (<https://www.nature.com/articles/s41586-021-03852-1/figures/6>). Therefore, the claim is utterly not true. The authors need to re-examine how the dataset is analysed, scaled and plotted to avoid misinterpretation.

2) The authors further concluded that "stem cells became the major source of α -defensins, REG3A and REG3G in the human proximal SI" citing Fig. 2e, f and S2e. First, the Fig. S2e showed a bar plot of distal SI (normalized count sum? average?), so it remains unclear how this plot provided evidence for the claim about proximal SI. Second, by summing up transcript counts from all the stem cells or TA cells, these cells might have more AMP transcripts (some of the AMPs or all of the AMPs?) than Paneth cells in proximal SI. However, this does not provide any functional implication of whether stem cells are the source of AMPs. Can AMPs be translated, cleaved and secreted by stem cells? This remains highly questionable.

3) With the findings of "human and mouse ileal epithelium ubiquitously expressed SI-specific AMPs in various epithelial lineages" and "small intestinal T and B cells expressed high levels of SI-specific AMPs", authors further conclude that the regional AMPs expression is regulated by environmental signals rather than region-specific cell differentiation. First, the ileal epithelium does not ubiquitously express the SI-specific AMPs (LYZ, DEFA5/6, REG3A/G, PLA2G2A, ITLN2, PRSS2), not even supported by the figures from the manuscript itself. In fact, these genes are regularly used as unique cell markers to define ileum Paneth cells in various scRNA-seq reports either from human tissues or organoids ([doi.org:10.1084/jem.20191130](https://doi.org/10.1084/jem.20191130), [doi.org:10.1016/j.cell.2020.12.016](https://doi.org/10.1016/j.cell.2020.12.016), [10.1016/j.stem.2022.08.002](https://doi.org/10.1016/j.stem.2022.08.002)). Second, indeed there are some AMP transcripts from subset of immune cells, but not at high levels. For instance, in Fig. 2i, 2k and S2g, both T cells and B cells showed almost no expression of ITLN2 and PLA2G2A. The authors should be rigorous in describing results and drawing conclusions, avoid overstating. Furthermore, the argument of regional AMPs regulated by environmental signals rather than region-specific cell differentiation is also not sustainable. It could be both. Regionally specific environmental signals, such as cytokines, can also trigger specific cell differentiation, resulting in AMPs expression (IL22 regulates Paneth cell differentiation in ileum organoids resulting in Defensins expression, meanwhile IL22 can also induces universal expression of certain AMPs such as REG1A, REG1B, [doi.org:10.1016/j.stem.2022.08.002](https://doi.org/10.1016/j.stem.2022.08.002)).

4) The authors performed gene regulatory network inference based on co-expression and highlighted that the bile acid receptors are the upstream regulators of AMPs. In Fig. 3a, can authors explain why bile acid receptors of certain lineage trees are highlighted, but not in others such as the lineages of enterocytes or lineages in prenatal tissues? Based on Fig. 3h, NR1H4 also showed equivalent expression levels in vast majority of enterocytes as Paneth cells. Can authors explain why the "BATFs-AMPs" is cross-lineage but not taking enterocytes into account? In fact, NR1H4 downstream target FABP6 has been identified as an enterocyte marker for bile acid uptake.

5) The manuscript profiled gene expression in Nr1h4 knockout mice, demonstrating a mild

decrease in a few AMPs. The authors should present all the AMPs gene expression mentioned in the previous sections in a heatmap. By using whole tissue for bulk RNA-seq, the gene expression readouts are significantly affected by other physiological factors. For instance, Nr1h4 total knockout mice showed significantly higher intestinal permeability, which could lead to more immune inflammation response. Instead of tissue RNA-seq, the authors can culture organoids from these mouse lines to have a pure readout from epithelium. Meanwhile, there are different readouts among intestinal conditional knockout, total knockout and liver knockout. The authors suggest that there is an antagonistic effect between intestine and liver Nr1h4 knockout. This is pure speculation without any data support. Since the authors continues to demonstrate the ligand-independent mechanisms of BATF regulation, how such antagonistic effect can be achieved between gut-liver axis without involving ligand activation? Overall, the tissue bulk RNA-seq data does not give much information for the question that review asked. To demonstrate the direct regulatory mechanism, using mouse organoid model with gene knockout will be a better strategy.

6) In Figure 6, the authors demonstrate an expression correlation between DEFA5/6 and some of the BATFs during fetal development and IBD versus healthy tissues. Such correlation cannot be used to draw a causal relationship. Can authors provide a cell percentage count of Paneth cell clusters in different conditions, like Fig. 6n? If the Paneth cell number has been shifted in different condition, then the genes expressed in Paneth cells will probably demonstrate the same co-expression pattern. To directly validate the ligand-independent regulatory mechanism of the proposed "BATFs-AMPs" axis in human, a gene knockout and ligand treatment can be performed in human intestinal organoids.

Other points:

1) The authors should consider changing the color palette in some of the plots, the small-scale transition from green to blue is not reader-friendly to tell the difference (such as Fig. 2c and 2d). To highlight the major message such as the gene expression comparison in difference cell clusters, authors should use violin plot rather than dot plot to present the expression distribution within each cell cluster. For cell clusters using similar color panel, the authors should label the name of the cell cluster next to the cell population in UMAP.

2) The scRNA-seq analysis presented in this study also highlights Paneth cells in colon and rectum tissues, with only 7 cell counts in colon and 1 cell count in rectum. The commonly accepted notion in the field is that there are no defined Paneth cells in large intestine under healthy condition, as shown in Fig. 2g. The authors should consider rename these cell clusters.

3) The authors used DEFA5 and LGR5 staining to define DEFA5+ stem cells and DEFA5+ non-stem cells in Fig. 2g and Fig. S2d. DEFA5 can be secreted and diffused in the tissue (like the pictures in Fig. 3e). LGR5 is also cell membrane localized. In fact, the staining does not look conclusive due to the unspecific signals in most of the crypt cells. Since Paneth cells and stem cells are next to each other, the co-staining is not reliable to determine double-positive cells. Can authors use RNAscope images to define cells expressing both LGR5 and Paneth cell AMPs? And the authors should provide the information of all the antibodies used in this study, which is currently missing.

4)

Typo in Fig. 3k: "State3" should be "Stat3".

Type at Line 171, "detection of BATFs" should be "deletion/disruption"?

Reviewer #3 (Remarks to the Author):

Summary: The authors demonstrate that regional heterogeneity of intestinal antimicrobial peptides (AMPs) is regulated by bile acids (BAs) through binding of BA transcription factors (BATFs) to AMP gene promoter regions. Further investigation revealed that chromatin accessibility at AMP gene promoters differs by region and correlates with the differential regional expression of those AMPs. Using single cell transcriptomics, the authors describe spatial, temporal, and cell type-specific differences in AMP expression in the small intestine. Bioinformatic analyses supported by

in vivo and organoid experiments indicate that signals from the organ environment play a stronger role than cell lineage in determining AMP expression capacity. ChIP-seq confirmed the presence of BATF binding sites at promoters of AMP genes, demonstrating that BATFs regulate AMP expression through a ligand-independent mechanism. scATAC-seq revealed that the degree of chromatin accessibility in specific regions dictates AMP expression. These findings resolve prior conflicting reports regarding the contribution of BATFs in the regulation of intestinal AMP expression with additional significance to human disease provided by transcriptomics in CD patients.

Overall Comments: This manuscript relies on a carefully executed, systematic approach to define a detailed mechanism by which BATFs regulate regional expression of AMPs in the gut. This includes thorough multi-omics analyses accompanied by some in vivo validation of the proposed mechanism using organoids and mouse models. The data supporting chromatin accessibility and BATF binding motifs at AMP promoter sites are convincing, as are the changes in AMP expression in organoids with conditional BATF knockouts. This study provides a regional map of AMP producing cells and further extends the cellular heterogeneity involved in the production of intestinal AMPs. The authors elegantly resolved the question of whether FXR signaling contributed to PC defects by demonstrating that BAs failed to affect alpha-defensin expression, but rather functioned to inhibit differentiation of secretory cells. However, inclusion of immunostaining would further support this bioinformatic observation.

Major Concerns:

1. Line 212: Stating that the authors will address the “mechanism by which DCA causes Paneth cell deficiency” is inaccurate since no data was shown to indicate that there is a change in PC number, only in PC-AMPs.
2. Line 223: Transcriptomic data should be confirmed in vivo and/or in an organoid model to demonstrate that FXR signaling reduces secretory lineages to strengthen the stated conclusion.
3. Line 283-285: The authors’ statement that “disordered BATFs-AMPs axis rather than WNT-AMPs or TLR-AMPs axis is involved in the collapse of intestinal antibacterial ability during CD” is an overstatement if only based on gene expression alone.
4. The authors hypothesize that the regional heterogeneity of SI AMPs helps shape the microbial communities in each region and propose in the discussion that targeting BATF-dependent AMP expression in specific regions may be an effective strategy to restore homeostasis of the microbiome. Since these are interesting points and are highly relevant in the context of IBD, further discussion of these ideas would be beneficial to framing the impact of these authors’ findings.

Minor Concerns:

1. Several typos throughout text, figures, and figure legends.
2. Figure panels should be referred to in order in the text (Fig 6)
3. In Fig. 4f, the treatment conditions are not related to each other; therefore, the connecting line between treatments is unnecessary.
4. In Fig. 6j, it is unclear whether all genes exhibit the same statistical significance or if the statistics only apply to DEFA5.

Dear Reviewers:

We would like to thank you for your careful reading, helpful comments, and constructive suggestions. These comments are all valuable for revising and improving the presentation of our manuscript, as well as the important guidance for our researches. We have studied all comments carefully and have made correction which we hope to sufficiently answer reviewers' concerns. The main corrections in the study and the responds to the reviewer' s comments are as follows.

REVIEWER COMMENTS

Reviewer #1 (Remarks to the Author)

The intestine is an organ for the absorption of nutrients which exists extensively host-microbe interaction. The organ has evolved diverse mechanisms against microbial invaders, including antimicrobial peptides. However, the heterogeneity in cell composition or function among intestine segments is not well resolved. In this manuscript, Wang et al. provided an intestinal cell landscape by integrating scRNA-Seq datasets produced by themselves or published before and described regional heterogeneity and cell type specificity of intestinal AMPs. Then they predicted that the BATFs were the upstream regulators of AMPs and chromatin accessibility determines the potential of BATFs and shapes the regional heterogeneity of intestinal AMPs. Finally, the authors infer the BATFs-AMPs axis participated in the establishment of intestinal antimicrobial barriers in fetuses and that the defects of BATFs mediate disruption of AMPs during Crohn's disease.

Overall, the topic is interesting and the content and quality of the manuscript are satisfactory. However, some points could be confirmed carefully. For example, the integration from different sources should set more strict filter criteria. More evidence should be provided to support the conclusion that immune cells also expressed high level of AMPs.

It is very interesting and efficient that the authors integrated amounts of published omics datasets to support the conclusion of the manuscript. But the filter and analysis should be described in more details. Necessary quality controls are often missed in integration. Moreover, the showing in the manuscript is very confusing in this version, FPKM, TPM, mRNA counts, and values with log-transformed...

Before the manuscript could be accepted by the journal, some suggestions could be considered.

Reply: Thanks for your valuable comments. We really appreciate your efforts in reviewing our manuscript. We have revised the manuscript accordingly. Our point-by-point responses are detailed below. The revised portion are marked in red in each response.

Major concern #1-1 The authors did not set strict filter criteria for the integration of intestine cells from a different source. The integration often faced the problem of over-integration or not enough, the negative and positive control should be set. For example, cells from the liver could be used as a negative control.

Reply: Thanks for your suggestion. In the revised manuscript, we use the scATAC-seq dataset containing liver samples (GEO: GSE184462) to display a UMAP plot of liver, ileum, and colon epithelial cells (Fig. 5d in the revised manuscript). The UMAP plot shows that epithelial cells from the liver are well separated from those from the intestine.

Fig. 5d | UMAP plot shows the epithelial cells from the liver, colon and ileum.

Major concern #1-2 And cells with the same identity should be integrated into the same cluster derived from different patients. These controls were not shown in the manuscript.

Reply: Thanks for your suggestion. As you mentioned, in studies using single-cell sequencing technology, it is common to display the distribution of samples in different cell clusters within UMAP or t-SNE embeddings to confirm that cells are appropriately integrated. However, due to the large number of samples integrated in our work (371 samples), the differences in color representing different samples projected onto the UMAP were too small to distinguish. Therefore, we did not show this result in the original manuscript. In the revised manuscript, we use bar plots to show the contributions of different samples in various cell clusters in order to confirm that each cell cluster is composed of cells from different samples (Supplemental Fig. 2a-d in the revised manuscript).

Supplemental Fig. 2a-d | Bar plots showing the contributions of different samples in various cell clusters in postnatal human intestine(a), prenatal human intestine (b), human iPSC-derived organoid (c) and mouse intestine (d).

In addition, we add detailed descriptions of the dataset quality control, integration, and normalization in the *Supplemental methods* section of the revised manuscript.

Line 580-592 in revised manuscript:

Dataset integration

This work incorporates numerous open-source scRNA-seq datasets, which may confound biological effects when analyzing inter phenotypic differences due to batch effects in sample processing, sequencing pipelines between different sources. To alleviate the impact of batch effects, we referred to the batch effect solution of a high-quality single-cell meta-analysis study, which integrated the macrophages from 14 organs of 41 public datasets⁵⁴. Specifically, we adopted three steps that introduced additional limitations while mitigating batch effects. Firstly, we only integrated datasets that were open source in the form of raw counts by using 10x Genomics single-cell sequencing technology, excluded datasets that were subjected to normalization of expression matrix, and datasets with less than 20,000 genes. Secondly, when merging the gene expression matrices of different datasets, only the genes included within all datasets were taken forward. Thirdly, the total counts were normalized to 10,000 reads per cell, so that counts become comparable among cells from different datasets.

Major concern #1-3 In addition, the marker genes for cell identity annotation in Fig. S1 should be given the corresponding references or sufficient evidence for identity definition.

Reply: Thanks for your suggestion. We add the corresponding references to the marker genes used for cell annotation in the *Supplemental methods* of the revised manuscript. Apart from *PI3*⁺ enterocyte and *IGKC*⁺ enterocytes, which are annotated by us for the first time based on their unique gene expression patterns.

Line 637-671 in revised manuscript:

Re-clustering and differential gene expression analysis (scanpy.tl.rank_genes_groups method, method='wilcoxon') were performed on each epithelial cell compartment to accurately identify cell types or subsets and characterize the differential genes of each subset (**Supplemental Fig. 1a-c**).

For the postnatal human epithelial cells, the results of preliminary re-clustering indicated that some lineages from different intestinal regions, such as absorptive enterocytes from proximal SI, distal SI, and LI were independent of each other in UMAP and cannot be grouped into a cluster. Therefore, we performed clustering and annotation on epithelial compartment of proximal SI, distal SI, and colorectum respectively, revealing 14 cell types and their proportional distribution and differential genes. For prenatal human epithelial cells, we first integrated fetal scRNA-seq data with postnatal data on defined cell types and performed dimensionality reduction, clustering, and visualization to enable label transfer of fetal cells with the help of postnatal epithelial cell identity definitions. Epithelial lineage cells shared between pre- and postnatal data were stem cells (*LGR5*, *ASCL2*, *SMOC2*, *RGMB*)^{15,22,24}, transit-amplifying cells (TA; *MKI67*, *TOP2A*)^{15,22,24}, Paneth cells (*DEFA5*, *DEFA6*, *REG3A*, *PRSS2*)¹⁵, goblet cells (*CLCA1*, *FCGBP*, *ZG16*, *MUC2*)^{15,22,24}, *BEST4*⁺ enterocytes (*BEST4*, *OTOP2*)^{15,22}, immature enterocytes (*OLFM4*, *DMBT1*)^{15,22}, *APOA1*⁺ enterocytes (*APOA1*, *ANPEP*, *FABP2*)¹⁵, *CA1*⁺ colonocytes (*CA1*, *CA2*, *SLC26A3*)^{15,22}, *PGC*⁺ mucous cells (*PGC*, *TFF2*, *MUC6*)⁶¹, enteroendocrine cells (*CHGA*, *CHGB*, *NEUROD1*)^{15,22,24}, microfold cells (*CCL20*, *CCL23*, *MIA*)^{15,22,24}, Tuft cells (*IRAG2*, *SH2D6*, *AZGP1*)^{15,22,24}, *PI3*⁺ enterocytes (*PI3*, *CD74*, *LCN2*), *IGKC*⁺ enterocytes (*IGHA1*, *IGHA2*, *JCHAIN*, *IGKC*) (**Fig. 1c and Supplemental Fig. 1a**). It should be noted that we also identified a small number of cells expressing Paneth cell markers in colon and rectum tissues derived from healthy individuals (7 cells in the colon, 1 cell in the rectum). Given that the right colon does normally have a small number of Paneth cells^{62,63}, as well as the expression all Paneth cell markers (*CCL24*, *DEFA5*, *DEFA6*, *REG3A*) in these cells, they were retained and annotated as Paneth cells. The single cell in the rectum only expressed *DEFA6* was removed.

For the human intestinal organoid compartment, the cells were preliminarily divided into gut-like cells and lung/esophagus-like cells based on the marker genes. Then, the gut-like cells were clustered into 7 clusters and annotated as stem cells^{15,22,24}, immature enterocytes^{15,22}, enterocytes¹⁵, *BEST4*⁺ enterocytes^{15,22}, *PGC*⁺ mucous cells⁶¹, enteroendocrine cells^{15,22,24}, and goblet cells^{15,22,24} the lung/esophagus-like cells were clustered into 2 clusters, basal like cells (*KRT4*, *TP63*)²⁴ and ciliated like cells (*FOXJ1*, *PIFO*)²⁴ (**Fig. 1c and Supplemental Fig. 1b**).

For the mouse intestinal epithelial cells, we respectively performed clustering and annotation on epithelial compartment of ileum and colon tissue, revealing 12 cell

types and their proportional distribution and differential genes^{20,64}(Fig. 1d and Supplemental Fig. 1c).

Major concern #2 In lines 112-123, the authors announced that the immune cells still expressed high levels of SI-specific AMPs. They should give more evidence. In my opinion, three points could be considered. 1) Sequencing reads from immune gene and AMP genes should be counted in the same cells but not in defined clusters to exclude the possibility of mis-annotated cells. 2) The immune cells in other organs or tissues should be checked to confirm whether the co-expression is specific in the small intestine. 3) The co-immunofluorescent assay for immune and AMPs in the same cell.

Reply: Thanks for your suggestion. In the revised manuscript, we have provided more evidence for the expression of SI-specific AMPs in small intestinal immune cells according to your suggestions, including:

1) Scatter plots were used to demonstrate the co-expression of the immune lineage marker genes *CD3D*, *CD79A* and *DEFA5* (a representative SI-specific AMP) in T cells and B cells in the proximal small intestine (SI), distal SI, and large intestine (LI), confirming the expression of immune marker genes and AMP in the same cell (Supplemental Fig. 6e, f in the revised manuscript).

Supplemental Fig. 6e, f | Scatter plots showing the co-expression of the immune lineage marker genes *CD3D*/*CD79A*, and *DEFA5*.

2) We analyzed the expression of *DEFA5* in immune cells of 44 organs using a scRNA-seq dataset of immune cells from 44 postnatal human tissues (Tabula Sapiens atlas)⁴⁶. The results indicate that, consistent with our internal discovery dataset, *DEFA5* is highly expressed in the immune cells of SI and is barely expressed in other tissues (Supplemental Fig. 6g-k in the revised manuscript).

Supplemental Fig. 6g-k | *DEFA5* is highly expressed in the immune cells of small intestinal tissue but barely expressed in other organs.

3) Using CD3D and *DEFA5* antibodies, ileal T cells expressing *DEFA5* were

immunofluorescently stained, confirming the expression of DEFA5 in immune cells at the protein level (Fig. 2I in the revised manuscript).

Fig. 2I | Bowel sections from the human ileum were immunofluorescent stained for DEFA5 (red), CD3 (green), and DAPI (blue). Arrows point to T cells expressing DEFA5. Scale bars, 10 μ m. Staining repeated on two participants.

We also rewrite a description of immune cells in small intestine expressing SI-specific AMPs in the *Results* section as follow.

Line 131-142 in revised manuscript:

We were surprised to find that SI-specific AMPs, such as *DEFA5*, *DEFA6* and *REG3A* were widely expressed by even immune cells in the SI, especially in the distal SI (**Fig. 2j, k and Supplemental Fig. 6a-f**). A strict doublet detection procedure, which removed all droplets detected the epithelial cell markers *EPCAM* and *KRT8*, was performed for the T/ILCs and B/Plasma cell compartments to exclude the effect of doublets of epithelial versus immune cells. The remaining small intestinal T and B cells still expressed high levels of SI-specific AMPs (**Fig. 2k and Supplemental Fig. 6d**). To verify the above findings, we analyzed the expression of *DEFA5* in immune cells of 44 postnatal human organs using a external validation scRNA-seq dataset²⁶. The results indicate that, consistent with our internal discovery dataset, *DEFA5* is highly expressed in the immune cells of SI and is barely expressed in other tissues (**Supplemental Fig. 6g-k**). In addition, we also confirmed the expression of DEFA5 in ileal CD3D⁺ T cells at the protein level by immunofluorescence (**Fig. 2I**).

Major concern #3 The normalization and showing method for RNA-Seq are inconsistent. FPKM and histogram in Fig. 3i – 3k; TPM and histogram with points in Fig 4d,e; raw mRNA counts and line chart in Fig 4f; TPM with log-transformed and histogram with points in Fig 4g; TPM with log-transformed and points in Fig 4h. The same questions also existed in other figures. I suggested the authors use the consistent TPM or FPKM and histogram with points to show these results. And all the published

Fig. 3g, h; Fig. 4b, d-h; Fig. 5a, b

Major concern #4-1 The control also should be used for scATAC-seq. In fig 5d, cells from liver overlapped with cells from two other organs in the mass. The authors should explain the causes of these results.

Reply: Thanks for your comment. While examining the UMAP embedding visualization of the scATAC-seq data from the original manuscript, we found that the cells were subject to over-integration (Fig. 5d in the original manuscript). In the revised manuscript, we first sorted the epithelial cells from the ileum, colon, and liver (*EPCAM*⁺ in intestine datasets, and *ALB*⁺ in liver datasets). We then performed UMAP on these epithelial cells, and the results showed that the epithelial cells from the colon and ileum clustered together, while hepatocytes were distant from the intestinal epithelial cells (Fig. 5d in the revised manuscript).

Fig. 5d | UMAP embedding visualization of the scATAC-seq data from human

live, ileum and colon in the original (left) and the revised manuscript (right).

Major concern #4-2 Moreover, in my view, the cell from the liver shown in fig. 5d are less than other organs. Whether this cell number influences the peak levels shown in Fig. S4d (there is no Fig. S4d in the original manuscript, it should be Fig. S4b) and Fig 5e.

Reply: Thanks for your suggestion. As you mentioned, in the original manuscript, we used a scATAC-seq dataset (GSE184462) with far fewer liver samples (n=1) compared to ileum (n=3) or colon (n=7). In the revised manuscript, we randomly sampled epithelial cells from the ileum and colon to maintain a consistent number of cells (n = 2500 cells, respectively) included in the analysis across the three tissues. The results showed that the conclusions drawn by the reanalysis of sampled cells are consistent with those of all cells: We observed significant differences in the chromatin accessibility of intestinal AMP genes including SI-specific AMPs (*DEFA5*, *DEFA6*, *REG3G*, *REG3A*, *PRSS2*) and co-expressed or LI-specific AMPs (*LYPD8*, *WFDC2*, *PI3*) in different organs, with much lower chromatin accessibility of AMP genes in liver cells than that in ileal and colon cells (Supplemental Fig. 9b, c in the revised manuscript), indicating that chromatin accessibility limits the regulation of intestinal AMPs by BATFs within the liver at the pre-transcriptional level.

Supplemental Fig. 9b, c | Analyze of randomly sampled epithelial cells showing different chromatin accessibility of intestinal AMP genes in liver and intestine.

We also rewrite the description of scATAC-seq analysis in the *Results* section as follow.

Line 270-273 in revised manuscript:

An equal number of cells from each of the three organs was randomly sampled (n = 2500 cells, respectively) to avoid any bias due to differences in cell counts between

organs. After random sampling, the results were consistent with those obtained using all cells (**Supplemental Fig. 9b, c**).

Major concern #5 The author also discussed the relationship of the regional heterogeneity of AMPs and microbiota. Maybe the authors could explain this problem by re-analyzing the metagenome data of the intestine. Of course, this suggestion is not necessary.

Reply: Thanks for your suggestion. Although we believe that regional heterogeneous AMPs play a role in shaping the microbial communities in different parts of the gastrointestinal tract, we do not believe that a causal relationship can be inferred from correlation data for metagenomes and host-derived AMP expression alone in the absence of genetic perturbation (10.1016/j.immuni.2020.07.010). Therefore we did not add these data in the revised manuscript. The main reasons are as follows:

(1) Too many confounding factors: The adaptability of microbes in different regions of the host intestine is influenced by many factors other than host-derived AMPs, such as the distribution of nutrients from food sources like sugars, fats, and proteins, as well as microbial competition for survival, bacteriophages, and bile acids with antibacterial functions.

(2) Not all host-derived AMPs work by killing bacteria. For instance, LYPD8, an LI-specific AMP, inhibits the motility of flagellated bacteria by binding to their flagella, thus preventing pathogen invasion (10.1038/nature17406). Similarly, DEFA6, an SI-specific AMP, undergoes an orderly self-assembly process upon random binding to bacterial surface proteins, forming protofibrils and nanonets that envelop and entangle bacteria to prevent their invasion (10.1126/science.1218831).

We add the following to the *Discussion* section of the revised manuscript.

Line 391-397 in revised manuscript:

Given the obvious correlation between regional heterogeneity of AMPs and microbial communities,²⁶ there is a potential causal relationship between the spatial distributions of AMPs and microorganisms. The fetal data demonstrated that the regional AMP immunosurveillance is established within the sterile prenatal gut,¹⁷⁻¹⁹ implying that AMP regional heterogeneity is responsible for, rather than a consequence of, the regional heterogeneity of microbial community in the gut. In the future work, genetic perturbation models of AMP genes could be used to confirm the causal relationship between region-specific AMPs and microbial composition.⁴³

Minor points #1: The colors for cell type in Fig. 1b(i) are not distinguishable.

Reply: Thanks for your suggestion. We have improved the colors of Fig. 1b(i) as follow.

b Human regional intestinal cell atlas across spatial-temporal, homeostasis and disease

Fig. 1b(i) | Human regional intestinal cell atlas across spatial-temporal, homeostasis and disease with new colors.

Minor points #2: The figure legend for Fig 2a should state three figures were produced separately.

Reply: Thanks for your suggestion. We correct the figure legend for Fig. 2a as follow.

Line 955-956 in revised manuscript:

a Combined visualization of three UMAP embeddings of prenatal and postnatal human intestinal epithelium and human organoids.

Reviewer #2 (Remarks to the Author)

The manuscript presents an extensive characterization of human antimicrobial peptides (AMPs) expression in human and mouse gut. The authors performed trajectory inference based on single-cell transcriptomics to map the transcription factor regulatory landscape of the cell differentiation, and concluded that the bile acid-activated TFs are the upstream regulators of AMPs expression. The study further demonstrated that such regulation is ligand-independent, and highly depends on chromatin accessibility in a cell-type-specific manner. My concern relates to a few major claims that the authors made in the manuscript without providing conclusive evidences.

Reply: Thanks for your valuable comments. We really appreciate your efforts in reviewing our manuscript. We have revised the manuscript accordingly. Our point-by-point responses are detailed below. The revised portion are marked in red in each response.

Major concern #1-1 At Line 100 (Fig. 2c), the authors claimed that “LGR5+ stem cells and transit-amplifying (TA) cells in the human SI also expressed high levels of SI-specific AMPs (LYZ, DEFA5/6, REG3A/G, PLA2G2A, ITLN2, PRSS2).” But this is not upheld by the figures showed. In Fig. 2c, majority of the cell populations except Paneth cells from ileum has the minimal expression of these AMPs based on the scale bar. The stem cells and TA cells have the same low-level expression as all the enterocyte clusters based on the scale.

Reply: Thanks for your comment. We apologize for any confusion caused by the presentation of our original manuscript. In the revised manuscript, we have used t-SNE embedding and violin plots to re-present the original dot plot results. The newly added figures more accurately revealed the expression of SI-AMPs in different intestinal tissues and cell types, as follows:

(1) Since the number of colon cells is much greater than that of the proximal small intestine and ileum, we randomly sampled the same number of epithelial cells (10,000 cells) from the proximal SI, ileum, and LI, calculated the t-SNE embedding, and visualized the SI-specific AMPs *LYZ*, *DEFA5/6*, *REG3A/G*, *ITLN2*, *PRSS2* (Supplemental Fig. 3d in the revised manuscript), and stem cell marker *LGR5*, as well as TA cell marker *MKI67* (Supplemental Fig. 3e in the revised manuscript). *PLA2G2A* is an AMP co-expressed in SI and LI, not an SI-specific AMP. This error in the original manuscript has been corrected in the revised version. The results show:

- 1) *DEFA5*, *DEFA6*, *REG3A*, and *PRSS2* are highly expressed in Paneth cells. *DEFA5*, *DEFA6*, and *REG3A* are widely expressed in other ileum epithelial cell types, while *PRSS2* is expressed at a lower level in other ileum epithelial cell types.

- 2) *ITLN2* is specifically expressed in Paneth cells.

3) *REG3G* is mainly expressed in ileum stem cells, with a very low expression in Paneth cells.

4) Surprisingly, *LYZ* was widely expressed in the epithelial cells of proximal SI (Supplemental Fig. 3d in the revised manuscript), resulting in a higher overall level of *LYZ* than in the ileum even with fewer Paneth cells (Supplemental Fig. 3f in the revised manuscript). To validate this, we analyzed a published RNA-seq dataset (E-MTAB-1733) and compared the *LYZ* expression in human duodenum and ileum. The results indicate that, consistent with the scRNA-seq data, *LYZ* has a significantly higher expression level in the duodenum than in the ileum (Supplemental Fig. 3g in the revised manuscript).

Supplemental Fig. 3d | t-SNE embedding showing the expression of SI-specific AMPs (*DEFA5*, *DEFA6*, *REG3A*, *REG3G*, *LYZ*, *ITLN2*, *PRSS2*, and *REG3G*) in epithelial cells. The same number of cells (10,000 cells) were randomly sampled from the proximal SI, ileum, and LI.

Supplemental Fig. 3e | t-SNE embedding showing the expression of stem cell (*LGR5*) and TA (*MKI67*) markers in epithelial cells. The same number of cells (10,000 cells) were randomly sampled from the proximal SI, ileum, and LI.

Supplemental Fig. 3f | Line chart showing the expression of three SI-specific AMPs (*LYZ*, *DEFA5*, *REG3A*) in proximal SI (Pro. SI), distal SI (Dis. SI) and LI in scRNA seq data.

Supplemental Fig. 3g | Bar plot of bulk RNA-seq data (E-MTAB-1733) showing that *LYZ* has a significantly higher expression level in the duodenum than in the ileum.

(2) We used violin plots to show the expression of SI-specific AMPs (*LYZ*, *DEFA5/6*, *REG3A/G*, *ITLN2*, *PRSS2*) in different epithelial cell types in the proximal and distal small intestine (Supplemental Fig. 3a, b in the revised manuscript). The results show:

1) In the proximal small intestine, stem cells have the second-highest expression of *DEFA5*, *DEFA6*, *REG3A*, *ITLN2*, and *PRSS2* among cell types, after Paneth cells. The expression level of *REG3G* in stem cells is slightly higher than that in Paneth cells. However, the expression level of SI-specific AMPs in TA cells is indeed very low.

2) In the ileum, stem cells and TA cells express higher levels of *DEFA5*, *DEFA6*, and *REG3A*, and lower levels of *PRSS2*. Stem cells also express *ITLN2* and *REG3G* at lower levels.

Supplemental Fig. 3a, b | Violin plots showing the expression of SI-specific AMPs in different epithelial subsets in the proximal and distal small intestine.

We also rewrite a description of regional AMPs in the *Results* section as follow.

Line 101-107 in revised manuscript:

Notably, our data suggested that most SI-specific AMPs (i.e. *DEFA5/6*, *REG3A/G*, *PRSS2*, and *LYZ*) except *ITLN2*, were not restricted to Paneth cells in the SI (Fig. 2c, d and Supplemental Fig. 3a, b, d). *LGR5*⁺ stem cells and transit-amplifying (TA) cells in the human SI also expressed high levels of SI-specific AMPs (*DEFA5/6*, *REG3A/G*, *PRSS2*, and *LYZ*) (Supplemental Fig. 3a, b, d, e and 4c-e). Due to the difference in the number of *DEFA5*⁺ stem cells versus Paneth cells, stem cells serve as an important source of α -defensins, *REG3A* and *REG3G* in the human proximal SI, besides Paneth cells (Fig. 2e, f and Supplemental Fig. 3a).

Line 116-121 in revised manuscript:

Surprisingly, *LYZ* was widely expressed in the epithelial cells of proximal SI (Supplemental Fig. 3d), resulting in a higher level of *LYZ* than in the ileum even with fewer Paneth cells (Supplemental Fig. 3f and Fig. 2f). To validate this, we analyzed a published RNA-seq dataset (E-MTAB-1733)²⁵ and compared the *LYZ* expression in human duodenum and ileum. The results indicate that, consistent with the scRNA-seq data, *LYZ* has a significantly higher expression level in the duodenum than in the ileum (Supplemental Fig. 3g).

Major concern #1-2 Moreover, the staining pictures showed in Fig. 2g also indicated that the *DEFA5*-expressing cells are solely enriched in the crypt, not even in the TA zone.

Reply: Thanks for your comment. We apologize for not providing evidence at the

protein level for this conclusion in the original manuscript. In the revised manuscript, we added an ileal immunohistochemical image stained with the TA cell marker MKI67 (downloaded from <https://proteinatlas.org>) and compared it with our previous immunofluorescent and immunohistochemical images stained with DEFA5 (Supplemental Fig. 4e in the revised manuscript). The results indicate that the ileal TA region is also enriched with DEFA5.

Supplemental Fig. 4e | Immunofluorescent (left) and immunohistochemical (middle) images showing the expression of DEFA5 in TA cells. Immunohistochemical (right, from <https://proteinatlas.org>) image showing the location of TA cells in the crypt by MKI67 staining. Scale bars, 50 μ m in immunofluorescent image and 100 μ m in immunohistochemical images.

Major concern #1-3 Most importantly, this manuscript included one of the recent dataset from “Cells of the human intestinal tract mapped across space and time (<https://doi.org/10.1038/s41586-021-03852-1>)”, which clearly showed the specificity of these AMPs in Paneth cells in their original analysis (<https://www.nature.com/articles/s41586-021-03852-1/figures/6>). Therefore, the claim is utterly not true. The authors need to re-examine how the dataset is analysed, scaled and plotted to avoid misinterpretation.

Reply: Thanks for your comment. We perform a separate analysis on the dataset you mentioned (<https://doi.org/10.1038/s41586-021-03852-1>, dataset available at gutcellatlas.org), splitting it by developmental time and anatomical region, and creating dotplots for SI-specific AMPs to confirm that the conclusions of our study can be fully replicated in this dataset: postnatal ileal epithelial cells broadly express *DEFA5*, *DEFA6*, and *REG3A*, and postnatal proximal SI *LGR5*⁺ stem cells also express *DEFA5*, *DEFA6*, and *REG3A* (Supplemental Fig. 5a, b in the revised manuscript). What led this study (doi.org/10.1038/s41586-021-03852-1) to suggest that Paneth cells were the only source of SI-specific AMPs is that the fetal epithelial

data and postnatal LI epithelial data they used masked SI-specific AMPs expression in non-Paneth epithelial cells in the postnatal ileum. Please note that in our reanalysis, we used the cell annotations provided by the authors of that study, thus ruling out potential discrepancies due to different cell annotations.

Supplemental Fig. 5a, b | Dot plots showing the expression of SI-specific AMPs in different epithelial subsets in prenatal and postnatal intestine.

Line 112-115 in revised manuscript:

Additionally, we found out the reason why previous scRNA-seq studies defined Paneth cells as the sole source of SI-specific AMPs, such as *DEFA5/6* and *REG3A*.¹⁵ This is because the fetal epithelial data and postnatal LI epithelial data they used

masked SI-specific AMPs expression in non-Paneth epithelial cells in the postnatal ileum (Supplemental Fig. 5a, b).

Major concern #2-1 The authors further concluded that “stem cells became the major source of α -defensins, REG3A and REG3G in the human proximal SI” citing Fig. 2e, f and S2e. First, the Fig. S2e showed a bar plot of distal SI (normalized count sum? average?), so it remains unclear how this plot provided evidence for the claim about proximal SI.

Reply: Thanks for your comment. Upon your reminding, we are aware that we have incorrectly used the image of the distal SI (Fig. S2e in the original manuscript), and we apologize for any inconvenience caused. In the revised manuscript, we provide the following three sets of data to demonstrate that stem cells are the main source of *DEFA5*, *DEFA6*, *REG3A*, and *REG3G* in the proximal SI:

(1) We provide violin plots of SI-specific AMP expression in epithelial cells of the proximal SI, which show that, in terms of gene expression intensity, proximal SI stem cells express high levels of *DEFA5*, *DEFA6*, *REG3A*, only second to proximal SI Paneth cells, while the expression of *REG3G* is even slightly higher than that of Paneth cells (Supplemental Fig. 3a in the revised manuscript).

Supplemental Fig. 3a | Violin plots showing the expression of SI-specific AMPs in different epithelial subsets in the proximal SI.

(2) We plotted barplots of the proportions of Paneth cells and stem cells in the epithelial cells of the proximal and distal SI (Fig. 2f in the revised manuscript). The results show that the number of stem cells in the proximal SI is much greater than that of Paneth cells. Therefore, in terms of the total expression of *DEFA5*, *DEFA6*, *REG3A*, and *REG3G*, proximal SI stem cells may contribute more than Paneth cells.

Fig. 2f | Bar plots showing the proportions of Paneth cells and stem cells in the epithelial cells of the proximal and distal SI.

(3) By plotting Wishbone graphs of the proximal and distal SI, we found a co-expression trend of stem cell marker *LGR5* with SI-specific AMPs, such as *DEFA5*, in the proximal SI (Fig. 2e in the revised manuscript). This further confirms that stem cells in the proximal SI are the main source of *DEFA5*, *DEFA6*, *REG3A*, and *REG3G*.

Fig. 2e | Wishbone graphs of the proximal and distal SI showing a co-expression trend of stem cell marker *LGR5* and SI-specific AMPs.

Additionally, we revise the original statement "stem cells became the major source of α -defensins, *REG3A* and *REG3G* in the human proximal SI" to "stem cells serve as an important source of α -defensins, *REG3A* and *REG3G* in the human proximal small intestine, besides Paneth cells" to avoid any potential exaggeration.

Line 105-108 in revised manuscript:

Due to the difference in the number of *DEFA5*⁺ stem cells versus Paneth cells, stem cells serve as an important source of α -defensins, *REG3A* and *REG3G* in the human proximal SI, besides Paneth cells (Fig. 2e, f and Supplemental Fig. 3a).

Major concern #2-2 Second, by summing up transcript counts from all the stem cells or TA cells, these cells might have more AMP transcripts (some of the AMPs or all of the AMPs?) than Paneth cells in proximal SI. However, this does not provide any

functional implication of whether stem cells are the source of AMPs. Can AMPs be translated, cleaved and secreted by stem cells? This remains highly questionable.

Reply: Thanks for your comment. In the revised manuscript, we add RNAScope *in situ* hybridization experiments for *LGR5* and *DEFA5* (Fig. 2g and Supplemental Fig. 4d in the revised manuscript), which, together with the co-staining immunofluorescence images of *LGR5* and *DEFA5* (Fig. 2h in the revised manuscript), demonstrate that not only is *DEFA5* mRNA transcribed in SI stem cells, but *DEFA5* protein is also translated.

Fig. 2g and Supplemental Fig. 4d | RNAScope *in situ* hybridization of *DEFA5* (red), *LGR5* (green), and DAPI (blue) in bowel sections from the human ileum (left) and duodenum (right). Scale bars, 10 μ m. Staining repeated on two participants.

Fig. 2h | Bowel sections from the human intestine were immunofluorescent stained for DEFA5 (red), LGR5 (green), and DAPI (blue). Scale bars, 50 μm . Staining repeated on three participants.

We add a description of RNAScope experiments in the *Results* section as follow.

Line 108-112 in revised manuscript:

To verify the above findings, we employed RNAScope *in situ* hybridization experiments for *LGR5* and *DEFA5* (**Fig. 2g and Supplemental Fig. 4d**), which, together with the co-staining immunofluorescence images of *LGR5* and *DEFA5* (**Fig. 2h**), demonstrate that not only is *DEFA5* mRNA transcribed in SI stem cells, but *DEFA5* protein is also translated.

Major concern #3-1 With the findings of “human and mouse ileal epithelium ubiquitously expressed SI-specific AMPs in various epithelial lineages” and “small intestinal T and B cells expressed high levels of SI-specific AMPs”, authors further conclude that the regional AMPs expression is regulated by environmental signals rather than region-specific cell differentiation. First, the ileal epithelium does not ubiquitously express the SI-specific AMPs (*LYZ*, *DEFA5/6*, *REG3A/G*, *PLA2G2A*, *ITLN2*, *PRSS2*), not even supported by the figures from the manuscript itself.

Reply: Thanks for your comment. We apologize for any confusion caused by the presentation of our original manuscript. In the revised manuscript, we have used t-SNE embedding, violin plots and ileal immunohistochemical image stained with the *DEFA5* to demonstrate that the ileal epithelium ubiquitously express the SI-specific AMPs (Supplemental Fig. 3b, d and Fig. 2i in the revised manuscript).

Supplemental Fig. 3d | t-SNE embedding showing the expression of SI-specific AMPs in epithelial cells from three intestinal regions.

Supplemental Fig. 3b | Violin plots showing the ubiquitous expression of SI-specific AMPs in the ileal epithelial cells.

Fig. 2i | Immunohistochemical image showing the ubiquitous expression of SI-specific AMPs in the ileal epithelial cells.

Major concern #3-2 In fact, these genes are regularly used as unique cell markers to define ileum Paneth cells in various scRNA-seq reports either from human tissues or organoids (doi.org:10.1084/jem.20191130, doi.org:10.1016/j.cell.2020.12.016, 10.1016/j.stem.2022.08.002).

Reply: Thanks for your comment. We understand that those genes mentioned by the reviewer have traditionally been considered as specific markers of Paneth cell. However, our work aims to demonstrate that these genes, although highly expressed in Paneth cells, are also broadly expressed in various types of epithelial cells in the postnatal ileum. We note that among the three studies mentioned by the reviewer, only the first study (doi.org:10.1084/jem.20191130) used postnatal human ileum epithelial cells. Therefore, we separately analyzed the dataset provided by this study (GEO: GSE125970). Using UMAP embedding, we show that, consistent with the results of our internal discovery set, *DEFA5*, *DEFA6*, *REG3A*, *REG3G*, *LYZ*, and *PRSS2* were also widely expressed in ileum epithelial cells (including stem cells, TA, intestinal

cells, goblet cells, intestinal endocrine cells, and Tuft cells) in this dataset, as shown below.

UMAP embedding from GEO: GSE125970 showing the broad expression of AMPs in the ileum epithelial cells.

In fact, there are instances of non-Paneth intestinal epithelial cells expressing SI-specific AMPs in the mouse small intestine as well. For example, Elisabeth E. L. Nyström and colleagues found that goblet cells in the mouse ileum also expressed *Defa24* and *Lyz1* (10.1126/science.abb1590, Fig. S6C).

Of note, when we re-analyzed this dataset (GSE125970), we found that the Paneth cells had been incorrectly annotated (doi.org:10.1084/jem.20191130, Fig. 1D). They used *SPIB*, *CA7*, *BEST4*, *CA4*, and *LYZ* to define ileal Paneth cells, but this actually represented *BEST4+* enterocytes^{17,24}. The proportion of true Paneth cells in the dataset GSE125970 (0.3%) was much lower than that in our own samples (9.4%) and other datasets, such as gutcellatlas.org (11.2%). This discrepancy may be due to the fact that the samples used in the GSE125970 dataset were not from healthy individuals, but from adjacent tissues of patients with intestinal tumors.

Major concern #3-3 Second, indeed there are some AMP transcripts from subset of

immune cells, but not at high levels. For instance, in Fig. 2i, 2k and S2g, both T cells and B cells showed almost no expression of *ITLN2* and *PLA2G2A*. The authors should be rigorous in describing results and drawing conclusions, avoid overstating.

Reply: Thanks for your suggestion. In the revised manuscript, we have redrawn the images of small intestinal immune cells expressing SI-specific AMPs. Images of *ITLN2* and *PLA2G2A* have been removed, while images of *DEFA5* and *REG3A*, which have higher expression levels, have been retained (Fig. 2k and Supplemental Fig. 6d in the revised manuscript).

Fig. 2k and Supplemental Fig. 6d | The expression of *DEFA5* and *REG3A* in SI immune cells.

We have also rewritten the description of AMP expression in small intestinal immune cells in the revised manuscript to avoid any exaggeration of the conclusions, as follows:

Line 130-133 in revised manuscript:

Furthermore, we were surprised to find that SI-specific AMPs, such as *DEFA5*, *DEFA6* and *REG3A* were widely expressed by even immune cells in the SI, especially in the distal SI (Fig. 2j, k and Supplemental Fig. 6a-f).

Major concern #3-4 Furthermore, the argument of regional AMPs regulated by environmental signals rather than region-specific cell differentiation is also not sustainable. It could be both. Regionally specific environmental signals, such as cytokines, can also trigger specific cell differentiation, resulting in AMPs expression (IL22 regulates Paneth cell differentiation in ileum organoids resulting in Defensins expression, meanwhile IL22 can also induces universal expression of certain AMPs such as *REG1A*, *REG1B*, doi.org:10.1016/j.stem.2022.08.002).

Reply: Thanks for your comment. First, we agree with your perspective that region-specific cell differentiation also plays a role in the regional patterns of AMPs. A great example is the differentiation of Paneth cells, which occurs almost exclusively in the small intestine under homeostatic conditions. However, in the context of our study, we believe that region-specific epigenetic regulatory signals (i.e., chromatin accessibility) are more important for the regulation of regional AMPs (especially SI-specific AMPs) than the specific cytokines on cell differentiation. The evidence is that previous studies have found that colonic Th17 cells express higher levels of IL22 than small intestinal Th17 cells under homeostatic conditions (10.1016/j.cell.2021.11.018, Fig. 1H). And, in reality, there are nearly no Paneth cells in the colon under healthy conditions (Fig. 1c in the revised manuscript).

Fig. 1c | The fraction of cells in different bowel segments. Paneth cells is barely seen in the healthy colon.

To further verify this, we used a published RNA-seq dataset (GEO: GSE190634) to analyze the expression of SI-specific AMPs after IL22 stimulation in human colonic organoids. The results showed that IL22 did not induce a significant upregulation of SI-specific AMPs (*DEFA5*, *DEFA6*, *REG3A*, *REG3G*, *ITLN2*). Moreover, although IL22 induced a significant upregulation of *REG1A* and *REG1B* in colonic organoids, neither *REG1A* nor *REG1B* are SI-specific AMPs. Both genes are widely expressed in the human colonic epithelium under homeostatic conditions, as follow.

Bar plots showing that IL22 did not induce a significant upregulation of SI-specific AMPs in human colonic organoids (GEO: GSE190634).

Violin plots showing the wide expression of *REG1A* and *REG1B* in human colonic epithelium under homeostatic conditions.

Major concern #4-1 The authors performed gene regulatory network inference based on co-expression and highlighted that the bile acid receptors are the upstream regulators of AMPs. In Fig. 3a, can authors explain why bile acid receptors of certain lineage trees are highlighted, but not in others such as the lineages of enterocytes or lineages in prenatal tissues?

Reply: Thanks for your comment. In Fig. 2 of the original manuscript, we have demonstrated the widespread expression of SI-specific AMPs in the postnatal ileum. Therefore, our focus in Fig. 3a of the original manuscript was on the highly expressed transcription factors in various cell types of the postnatal ileum. Postnatal colonic and fetal epithelium generally exhibit low expression of SI-specific AMPs (Fig. 2c and Fig. 6d in the revised manuscript). As a result, we did not highlight BATFs in the colonic and fetal epithelial lineages.

Major concern #4-2 Based on Fig. 3h, NR1H4 also showed equivalent expression levels in vast majority of enterocytes as Paneth cells. Can authors explain why the “BATFs-AMPs” is cross-lineage but not taking enterocytes into account? In fact, NR1H4 downstream target FABP6 has been identified as an enterocyte marker for bile acid uptake.

Reply: Thanks for your comment. Indeed, this question cannot be answered using the original manuscript's Fig. 3h alone, as the enterocytes of the ileum express *NR1H4* at similar levels to Paneth cells. In fact, the reviewer's question can be addressed with the scATAC-seq data. In the revised manuscript, we provided chromatin accessibility UMAP embeddings of the absorptive enterocyte markers *APOA1* and *APOA4* (Supplemental Fig. 9d in the revised manuscript). By comparing this data with the

UMAP embeddings of chromatin accessibility of *DEFA5* and *REG3A* (Supplemental Fig. 9e in the revised manuscript), it was observed that the chromatin accessibility for *DEFA5* and *REG3A* in absorptive enterocytes is closed. Therefore, even if enterocytes highly express *NRIH4*, they only express SI-specific AMPs at a low level.

Supplemental Fig. 9d | UMAP embedding showing the chromatin accessibility of *APOA1* and *APOA4* in the human ileum.

Supplemental Fig. 9e | UMAP embedding showing the chromatin accessibility of *DEFA5* and *REG3A* in the human ileum.

In the revised manuscript, we add the following content:

Line 279-284 in revised manuscript:

Furthermore, the single-cell chromatin accessibility map also reasoned why ileal enterocytes express *NRIH4* at similar levels to Paneth cells but relatively low levels of SI-specific AMPs (Fig. 3f): Chromatin accessibility of SI-specific AMPs also varies between different epithelial cell types within the same intestinal region, and the chromatin accessibility of SI-specific AMPs in *APOA1*+*APOA4*+ enterocytes is much lower than that in Paneth cells (Supplemental Fig. 9d, e).

Major concern #5-1 The manuscript profiled gene expression in *Nr1h4* knockout mice, demonstrating a mild decrease in a few AMPs. The authors should present all the AMPs gene expression mentioned in the previous sections in a heatmap.

Reply: Thanks for your suggestion. In the revised manuscript, we provide a heat map

of the changes in potential target genes downstream of *Nr1h4* (orthologs in mice) when *Nr1h4* is knocked out (Supplemental Fig. 8c in the revised manuscript). The results showed that most of the potential target genes were downregulated. In addition, we used Barplot in the main text to show the altered results of three major SI-specific AMPs, α -defensins (*Defa24*), C-type lectins (*Reg3b*, *Reg3g*) and lysozyme (*Lyz1*), upon *Nr1h4* deficiency (Fig. 3h in the revised manuscript).

Supplemental Fig. 8c | Heat map showing the expression of potential target genes of NR1H4 in *Nr1h4-fl/fl* (control) and *Nr1h4-intKO* mice.

Fig. 3h | Bar plots showing the expression of SI-specific AMPs (downstream of NR1H4) in WT and *Nr1h4* intestinal knockout mice.

Major concern #5-2 By using whole tissue for bulk RNA-seq, the gene expression readouts are significantly affected by other physiological factors. For instance, *Nr1h4* total knockout mice showed significantly higher intestinal permeability, which could lead to more immune inflammation response. Instead of tissue RNA-seq, the authors can culture organoids from these mouse lines to have a pure readout from epithelium. Overall, the tissue bulk RNA-seq data does not give much information for the question that review asked. To demonstrate the direct regulatory mechanism, using mouse organoid model with gene knockout will be a better strategy.

Reply: Thanks for your suggestion. First, we acknowledge that data from organoids of gene knockout mice, if consistent with *in vivo* data of *NR1H4*-deficient intestinal epithelium, would help confirm the direct regulatory relationship between BATFs and AMPs. However, we must clarify that the bulk RNA-seq data from the two knockout mice intestinal tissues shown in the original manuscript were reanalyzed from published datasets (GEO: GSE163157, GSE144978). Given the construction and breeding cycle of CRISPR-Cas9 gene-edited mice, we are unable to provide the

organoid data requested by the reviewer here.

Secondly, we need to point out that *Nr1h4* gene knockout organoids may also not be able to prove the direct regulatory relationship between BATFs and AMPs. Previous studies have demonstrated that defects or inhibition of *Nr1h4* can lead to abnormal activation of the WNT signaling pathway (10.1016/j.cell.2019.01.036, 10.1158/0008-5472.can-08-1791), which controls the differentiation of secretory epithelial cells, including Paneth cells (10.1152/ajpgi.00347.2013, 10.1038/ncb1240). Therefore, *Nr1h4* knockout may affect the expression of AMPs by altering the differentiation of organoids, making it impossible to determine the direct regulation of *Nr1h4* on AMPs.

The most reliable approach to address the aforementioned issue is a Paneth cell line that highly expresses both SI-specific AMPs and *Nr1h4*, in which *Nr1h4* is knocked out to explore the direct regulation of *Nr1h4* on SI-specific AMPs while not affecting cell differentiation. However, no Paneth cell line was available so far.

Major concern #5-3 Meanwhile, there are different readouts among intestinal conditional knockout, total knockout and liver knockout. The authors suggest that there is an antagonistic effect between intestine and liver *Nr1h4* knockout. This is pure speculation without any data support.

Reply: Thanks for your comment. We agree with the reviewer's view that the results that liver-specific knockout of *Nr1h4* upregulating SI-AMPs, intestinal-specific knockout of *Nr1h4* downregulating SI-AMPs, and simultaneous knockout of *Nr1h4* in both liver and intestine (i.e., systemic knockout) leading to SI-AMPs expression comparable to the control group are indeed not strong evidence to prove the antagonistic role of liver and intestinal *NR1H4*. Therefore, we remove these data, which does not pertain to the core argument of the paper (Core argument: Chromatin accessibility determines the potential of BATFs to regulate AMPs at the pre-transcriptional level, thus shaping the regional heterogeneity of AMPs), from the original manuscript.

Major concern #5-4 Since the authors continues to demonstrate the ligand-independent mechanisms of BATF regulation, how such antagonistic effect can be achieved between gut–liver axis without involving ligand activation?

Reply: Thanks for your comment. The liver-gut axis, which regulates the function of intestinal epithelial cells through bile acids, does not necessarily depend on ligand-dependent activation of bile acid nuclear receptors (FXR/NR1H4, LXR/NR1H3, VDR/NR1I1, PXR/NR1H2). For example, during inflammatory bowel disease, the excessive secretion of bile acids (CA) by the liver has been shown to impair the renewal of *Lgr5+* intestinal stem cells by inhibiting PPAR α , resulting in blocked fatty acid oxidation, which is independent of the activation of BATFs

(10.1016/j.stem.2022.08.008). In addition, besides BATFs (bile acid nuclear receptors), liver-secreted bile acids can also modulate the function of intestinal epithelial cells by stimulating their cell membrane receptor TGR5/GPBAR1 (10.1053/j.gastro.2020.05.067).

Major concern #6-1 In Figure 6, the authors demonstrate an expression correlation between DEFA5/6 and some of the BATFs during fetal development and IBD versus healthy tissues. Such correlation cannot be used to draw a causal relationship. Can authors provide a cell percentage count of Paneth cell clusters in different conditions, like Fig. 6n? If the Paneth cell number has been shifted in different condition, then the genes expressed in Paneth cells will probably demonstrate the same co-expression pattern.

Reply: Thanks for your suggestion. We understand the reviewer's concern that in Fig. 6, the correlation between BATFs and AMP expression levels during IBD onset and fetal development do not represent causation. Therefore, although we have provided evidence in Figures 3 and 4 of the original manuscript that BATFs modulate AMPs expression in a non-ligand-dependent manner, we still used speculative terms such as "may regulate" and "potential causal relationship" in summarizing the fetal development and IBD data in Fig. 6.

In order to avoid overinterpretation of the data, in the *Discussion* section of the revised manuscript, we emphasize that the changes of AMPs during development and intestinal inflammation should be studied in the future in epithelial-specific BATFs knockout mice, as follows:

Line 398-402 in revised manuscript:

Considering the consistent expression of BATFs and SI-specific AMPs during the fetal development and the progression of CD, the proposed BATFs-AMPs axis may be involved in the establishment of antimicrobial barriers of fetal gut and the defects in antibacterial ability of multiple lineages during CD, which needs to be further confirmed by models of epithelial-specific knockout of BATFs in the future.

In addition, based on the reviewer's suggestion, we plotted the cell proportion of Paneth cells and DEFA5+ non-Paneth cells at different stages of fetal development. The results show that the fraction of DEFA5 positive non-Paneth cells but not Paneth cells increased with fetal development and increased BATFs expression (Fig. 6g and Supplemental Fig. 10g in the revised manuscript). The unchanged proportion of Paneth cells is consistent with a previous study: David Fawkner-Corbett et al. investigated the proportional changes of different secretory epithelial cells during human fetal development and found that only goblet cells and enteroendocrine cells gradually increased with development (10.1016/j.cell.2020.12.016, Fig. S3E).

Fig. 6g | The expression level of *NR1H4* increased with fetal development.

Supplemental Fig. 10g | The fraction of *DEFA5* positive non-Paneth cells but not Paneth cells increased with fetal development.

We add a description of these data to the *Results* section of the revised manuscript.

Line 304-306 in revised manuscript:

More importantly, we observed that the proportion of *DEFA5*+ non-Paneth cells but not Paneth cells increased during development (**Supplemental Fig. 10g**), which further eliminated the effect of cell differentiation on the expression of SI-specific AMPs.

Major concern #6-2 To directly validate the ligand-independent regulatory mechanism of the proposed “BATFs-AMPs” axis in human, a gene knockout and ligand treatment can be performed in human intestinal organoids.

Reply: Thanks for your comment. We respectfully disagree with the reviewer's suggestion to implement BATFs knockout in human intestinal organoids or using organoids derived from gene knock out mice, as *NR1H4* deficiency has been shown to abnormally activate the WNT signaling pathway (10.1016/j.cell.2019.01.036, 10.1158/0008-5472.can-08-1791), which controls Paneth cell differentiation (10.1152/ajpgi.00347.2013, 10.1038/ncb1240). Therefore, *Nr1h4* knock out may affect the expression of AMPs by altering the differentiation of epithelial cells,

making organoids an inappropriate model to study the direct regulation of Nr1h4 on AMPs.

Other point #1 The authors should consider changing the color palette in some of the plots, the small-scale transition from green to blue is not reader-friendly to tell the difference (such as Fig. 2c and 2d). To highlight the major message such as the gene expression comparison in difference cell clusters, authors should use violin plot rather than dot plot to present the expression distribution within each cell cluster. For cell clusters using similar color panel, the authors should label the name of the cell cluster next to the cell population in UMAP.

Reply: Thanks for your comments. We have made the following adjustments in the revised manuscript according to your suggestions:

(1) Adjusted the color palette of Fig. 2c and Fig. 2d to the RdYlBu_r mode, transitioning from blue to red.

Fig. 2c, d | Dot plots with changed color palette.

(2) Displayed the expression of SI-specific AMPs in the proximal small intestine and distal small intestine, as well as LI-specific AMPs in the colorectum, using violin plots (Supplemental Fig. 2a-c in the revised manuscript).

Supplemental Fig. 2a-c | Violin plots showing the expression of SI-specific AMPs in the proximal and distal SI, and LI-specific AMPs in the LI.

(3) Annotated the cell cluster numbers next to the corresponding cell subpopulations in the UMAP plot (Fig. 2a in the revised manuscript).

Fig. 2a | Cell clusters are numbered in the UMAP embedding.

Other point #2 The scRNA-seq analysis presented in this study also highlights Paneth cells in colon and rectum tissues, with only 7 cell counts in colon and 1 cell count in rectum. The commonly accepted notion in the field is that there are no defined Paneth cells in large intestine under healthy condition, as shown in Fig. 2g. The authors should consider rename these cell clusters.

Thank you for your comment. In fact, a small number of Paneth cells are present in the right colon in healthy people, due to the fact that both the right colon and the ileum have developmental origins in the midgut (PMID: 8572559). Paneth cell metaplasia in the left colon and rectum is a pathological marker of IBD in humans (10.1046/j.1440-1746.2001.02629.x). In the revised manuscript, considering that only one *DEFA6*+ cell was found in the rectum and this cell did not express other human Paneth cell markers such as *DEFA5*, *CCL24*, and *REG3A* (Fig. 2c in the original manuscript), we removed the Paneth cell identified in the rectum. We provided the rationale for identifying Paneth cells in the healthy human colon in the revised manuscript's *Supplemental methods* section as follows:

Line 657-662 in revised manuscript:

It should be noted that we also identified a small number of cells expressing Paneth cell markers in colon and rectum tissues derived from healthy individuals (7 cells in the colon, 1 cell in the rectum). Given that the right colon does normally have a small number of Paneth cells^{62,63}, as well as the expression all Paneth cell markers (*CCL24*, *DEFA5*, *DEFA6*, *REG3A*) in these cells, they were retained and annotated as Paneth cells. The single cell in the rectum only expressed *DEFA6* was removed.

Other point #3 The authors used DEFA5 and LGR5 staining to define DEFA5+ stem

cells and DEFA5+ non-stem cells in Fig. 2g and Fig. S2d. DEFA5 can be secreted and diffused in the tissue (like the pictures in Fig. 3e). LGR5 is also cell membrane localized. In fact, the staining does not look conclusive due to the unspecific signals in most of the crypt cells. Since Paneth cells and stem cells are next to each other, the co-staining is not reliable to determine double-positive cells. Can authors use RNAScope images to define cells expressing both LGR5 and Paneth cell AMPs? And the authors should provide the information of all the antibodies used in this study, which is currently missing.

Reply: Thanks for your suggestion. In the revised manuscript, we add RNAScope *in situ* hybridization experiments for LGR5 and DEFA5 (Fig. 2g and Supplemental Fig. 4d in the revised manuscript), demonstrating that LGR5+ stem cells indeed express DEFA5. In addition, information on key reagents can be obtained from the `nr_report_summary` file submitted with this revision.

Fig. 2g and Supplemental Fig. 4d | RNAScope *in situ* hybridization of *DEFA5* (red), *LGR5* (green), and DAPI (blue) in bowel sections from the human ileum (left) and duodenum (right). Scale bars, 10 μ m. Staining repeated on two participants.

Other point #4

Typo in Fig. 3k: “State3” should be “Stat3”.

Typo at Line 171, “detection of BATFs” should be “deletion/disruption”?

Reply: Thanks for your suggestion. We have corrected these typos.

Reviewer #3 (Remarks to the Author)

Summary: The authors demonstrate that regional heterogeneity of intestinal antimicrobial peptides (AMPs) is regulated by bile acids (BAs) through binding of BA transcription factors (BATFs) to AMP gene promoter regions. Further investigation revealed that chromatin accessibility at AMP gene promoters differs by region and correlates with the differential regional expression of those AMPs. Using single cell transcriptomics, the authors describe spatial, temporal, and cell type-specific differences in AMP expression in the small intestine. Bioinformatic analyses supported by in vivo and organoid experiments indicate that signals from the organ environment play a stronger role than cell lineage in determining AMP expression capacity. ChIP-seq confirmed the presence of BATF binding sites at promoters of AMP genes, demonstrating that BATFs regulate AMP expression through a ligand-independent mechanism. scATAC-seq revealed that the degree of chromatin accessibility in specific regions dictates AMP expression. These findings resolve prior conflicting reports regarding the contribution of BATFs in the regulation of intestinal AMP expression with additional significance to human disease provided by transcriptomics in CD patients.

Overall Comments: This manuscript relies on a carefully executed, systematic approach to define a detailed mechanism by which BATFs regulate regional expression of AMPs in the gut. This includes thorough multi-omics analyses accompanied by some in vivo validation of the proposed mechanism using organoids and mouse models. The data supporting chromatin accessibility and BATF binding motifs at AMP promoter sites are convincing, as are the changes in AMP expression in organoids with conditional BATF knockouts. This study provides a regional map of AMP producing cells and further extends the cellular heterogeneity involved in the production of intestinal AMPs. The authors elegantly resolved the question of whether FXR signaling contributed to PC defects by demonstrating that BAs failed to affect alpha-defensin expression, but rather functioned to inhibit differentiation of secretory cells. However, inclusion of immunostaining would further support this bioinformatic observation.

Reply: Thanks for your valuable comments. We really appreciate your efforts in reviewing our manuscript. We have revised the manuscript accordingly. Our point-by-point responses are detailed below. The revised portions are marked in red in each response.

Major concern #1 Line 212: Stating that the authors will address the “mechanism by which DCA causes Paneth cell deficiency” is inaccurate since no data was shown to indicate that there is a change in PC number, only in PC-AMPs.

Reply: Thanks for your comment. Based on the bulk RNA-seq data of DCA-stimulated mouse small intestinal organoids, we proposed that DCA induced defects in secretory cells, such as Paneth cells and Goblet cells (Fig. 4i in the revised

manuscript). We agree with your concern that bulk RNA-seq data were insufficient to directly prove the fact that differentiation defect in Paneth cell is induced by DCA; DCA may only suppress the expression of Paneth cell-AMPs. To address this issue, we have added a scRNA-seq assay of mouse small intestinal organoids stimulated by DCA and DMSO (control) in the revised version to directly depict the changes in Paneth cells. The results are as follows:

(1) In 13,815 quality-controlled cells, we identified 8 cell subpopulations, including 2 undifferentiated lineages (Stem cell, TA), 3 absorptive enterocytes (Immature enterocyte, *Fabp1*+enterocyte, *Crbrd1*+enterocyte), and 3 secretory epithelial lineages (Tuft cell, EECs, Paneth-Goblet cell) (Fig. 4j in the revised manuscript). Notably, the specific markers of adult mouse Paneth cells, *Mmp7* and *Defa3*, were co-expressed with Goblet cell markers *Muc2* and *Fcgbp* in the same cells of the organoids (Fig. 4k, l in the revised manuscript), which is consistent with the findings of a previous scRNA-seq study on mouse small intestinal organoids (10.1016/j.cell.2018.10.008, Fig. S3F).

Fig. 4j | UMAP embedding of 8 epithelial cell subsets in mouse SI organoid (n = 13,815 cells).

Fig. 4k | Marker genes of adult mouse Paneth cell are co-expressed with goblet cell markers in intestinal organoids.

Fig. 4l | Marker genes of adult mouse Paneth cell are co-expressed with goblet cell markers in intestinal organoids.

(2) We analyzed the proportions of each cell type after DCA stimulation, and the results showed that, consistent with our hypothesis proposed using bulk RNA-seq, DCA induces differentiation defects in the secretory lineage and stem cells, leading to a decrease in the proportions of secretory epithelial cells (Paneth-Goblet cells), stem cells, and an increase in the proportions of absorptive enterocytes (Fig. 4m in the revised manuscript).

Fig. 4m | DCA treatment resulted in a decrease of secretory epithelial cells and stem cells, while an increase of absorptive epithelial cells.

We add a description of these data to the Results section of the revised manuscript.

Line 227-239 in revised manuscript:

ScRNA-seq experiments in mouse intestinal organoids stimulated with DCA and DMSO were employed to directly depict the changes in the proportion of epithelial lineages. In 13,815 quality-controlled cells, we identified 8 cell subpopulations, including 2 undifferentiated lineages (Stem cell, TA), 3 absorptive enterocytes (Immature enterocyte, *Fabp1*+ enterocyte, *Crbrd1*+ enterocyte), and 3 secretory epithelial lineages (Tuft cell, EECs, Paneth-Goblet cell) (Fig. 4j). Notably, the specific markers of adult mouse Paneth cells, *Mmp7* and *Defa3*, were co-expressed with Goblet cell markers *Muc2* and *Fcgbp* in the same cells of the organoids (Fig. 4k, l), which is consistent with the findings of a previous scRNA-seq study on mouse

small intestinal organoids³⁸. The changes in the proportions of each cell type after DCA stimulation indicated that DCA induced a decrease in the proportions of stem cells and secretory epithelial cells, and an increase in the proportion of absorptive enterocytes (Fig. 4m), confirming that DCA induced Paneth cell differentiation defects rather than suppressing the expression of Paneth cell-AMPs.

Major concern #2 Line 223: Transcriptomic data should be confirmed in vivo and/or in an organoid model to demonstrate that FXR signaling reduces secretory lineages to strengthen the stated conclusion.

Reply: Thanks for your comment. We realized that the description in the original manuscript may have caused ambiguity. Based on the observation that CDCA, CA, and DCA all significantly upregulated FXR downstream target genes (Fig. 4g in the revised manuscript), but only DCA downregulated secretory lineage markers and key genes in the WNT signaling pathway (Fig. 4i in the revised manuscript), we proposed the following viewpoint on line 223 of the original manuscript: “DCA **did not** cause Paneth cell defects through FXR signaling, but inhibited the differentiation of secretory lineages by inhibiting the Wnt signaling pathway, resulting in the down-regulation of Paneth cell-AMPs.” In other words, we believe that activating the FXR signaling pathway does not reduce the secretory lineage.

Fig. 4g, i | CDCA, CA, and DCA all induced upregulation of FXR target genes, but only DCA induced secretory lineage and stem cell defects.

In the revised manuscript, we have changed the way of presenting, as follows.

Line 243-245 in revised manuscript:

DCA suppresses the differentiation of the secretory lineages by inhibiting the Wnt signaling pathway, leading to the down-regulation of Paneth cell-AMPs, rather than through FXR signaling.

Additionally, in the revised manuscript, we used scRNA-seq to analyze the proportion of Paneth cells and other secretory lineage cells in organoids before and after DCA stimulation, confirming the conclusion that DCA reduces the differentiation of secretory lineages (Fig. 4m in the revised manuscript).

Fig. 4m | DCA treatment resulted in a decrease of secretory epithelial cells and stem cells, while an increase of absorptive epithelial cells.

Major concern #3 Line 283-285: The authors’ statement that “disordered BATFs-AMPs axis rather than WNT-AMPs or TLR-AMPs axis is involved in the collapse of intestinal antibacterial ability during CD” is an overstatement if only based on gene expression alone.

Reply: Thanks for your comment. We agree with your concern. In the revised manuscript, we removed the content of the WNT signaling pathway and TLRs signal transduction during Crohn's disease in Figure 6, and only retained the data of BATFs that validated at the protein level (Fig. 6k, l and Supplemental Fig. 11a-c in the revised manuscript) to avoid overstatement.

Major concern #4 The authors hypothesize that the regional heterogeneity of SI AMPs helps shape the microbial communities in each region and propose in the discussion that targeting BATF-dependent AMP expression in specific regions may be an effective strategy to restore homeostasis of the microbiome. Since these are interesting points and are highly relevant in the context of IBD, further discussion of these ideas would be beneficial to framing the impact of these authors’ findings.

Reply: Thanks for your suggestion. We add the following content to the *discussion* section of the revised manuscript.

Line 398-409 in revised manuscript:

Considering the consistent expression of BATFs and SI-specific AMPs during the fetal development and the progression of CD, the proposed BATFs-AMPs axis may be involved in the establishment of antimicrobial barriers of fetal gut and the defects in antibacterial ability of multiple lineages during CD, which needs to be further confirmed by models of epithelial-specific knockout of BATFs in the future. In addition, the stability of FXR is largely regulated by acetylation, while SIRT1 is a key regulator of FXR deacetylation⁴⁷, suggesting that inhibition of SIRT1 will increase

the stability of FXR protein. Recent studies have shown that intestinal epithelial cell-specific knockout of *Sirt1* leads to a significant upregulation of multiple SI-specific AMPs in the SI.⁴⁸ Collectively, SIRT1 may regulate the protein stability of FXR in epithelial cells through deacetylation to control the expression of intestinal AMPs. Therefore, SIRT1 inhibitors may become a promising approach to treat CD by restoring the homeostasis of AMPs and microbial communities.

Minor Concern #1 Several typos throughout text, figures, and figure legends.

Reply: Thanks for your suggestion. We have reviewed the full manuscript carefully, and corrected the problems such as grammar issues and order of figure panels.

Minor Concern #2 Figure panels should be referred to in order in the text (Fig 6)

Reply: Thanks for your suggestion. We have corrected these typos.

Minor Concern #3 In Fig. 4f, the treatment conditions are not related to each other; therefore, the connecting line between treatments is unnecessary.

Reply: Thanks for your suggestion. We have revised this figure accordingly (Fig. 4f in the revised manuscript).

Fig. 4f | Shown are the expression of predicted downstream SI-specific AMPs of BATFs in mouse small intestinal organoids treated with vehicle or 4 distinct bile acid pools, which varied based on the species (i.e., human (H) or mouse (M)), and the proportion of 12 α -hydroxylated bile acids (10% or 90%). Three representative replicates are shown.

Minor Concern #4 In Fig. 6j, it is unclear whether all genes exhibit the same statistical significance or if the statistics only apply to DEFA5.

Reply: Thanks for your suggestion. We revised the figure legend of Fig. 6j and 6k as

follows.

Line 1061-1062 in revised manuscript:

j, k SI-specific AMPs (j) and BATFs (k) in ileal epithelial cells decreased significantly with the progression of CD. Statistical significance applies to all five AMPs (*DEFA5*, *DEFA6*, *PRSS2*, *REG3G*, and *ITLN2*) and three BATFs (*NR1H4*, *NR1H3*, and *VDR*) in this figure.

REVIEWERS' COMMENTS

Reviewer #1 (Remarks to the Author):

The authors have answered all of my questions, and I have no further inquiries. I agree to accept the article.

Reviewer #2 (Remarks to the Author):

The authors have now answered the critiques on the bioinformatics analysis, and significantly improved the data presentation as requested. Though organoid experiments were not performed, considering the practical difficulties, the authors have addressed the major concerns by providing additional analysis on the published datasets. Potential overstatements have been adjusted in the discussion. Therefore, I believe the manuscript now can be accepted for publication. Below I attach my further elaboration.

My previous major critiques are: 1) The authors should provide biological evidences on the 'ubiquitous' expression of SI-specific AMPs in non-Paneth epithelial cells. 2) The authors should confirm the "BATFs-AMPs" axis identified by bioinformatics study in an organoid model.

For the first critique, the authors improved the data presentation to highlight the low-level AMP expression in non-Paneth cells in postnatal distal SI. And they further elaborated in the revised manuscript which AMPs are 'ubiquitous', which AMPs are highly restricted in Paneth cells (e.g. ITLN2). The authors also performed the RNA scope and staining as requested to confirm the expression of DEFA5 in LGR5+ cells in human ileum and duodenum.

For the second critique, the authors did not perform the organoid study (such as knockout BATF in organoids) to confirm the BATF-dependent AMP expression, but arguing that NR1H4 deficiency abnormally activates WNT signalling (previously proved in ApcMin/+ mice), and might affect Paneth cell differentiation, thus makes organoids an inappropriate model to study the direct regulation of NR1H4 on AMPs. First, it is not known whether the NR1H4 knockout indeed affects Paneth differentiation as it was not characterized in the previous report (10.1124/jpet.108.145409/). Since the authors have already proved above that there is SI specific AMP expression in LGR5+ cells, then it is reasonable to use organoids to profile the AMP expression upon NR1H4 knockout in LGR5+ cells specifically by FACS sort or scRNA-seq, which excludes the effect from potential bias of secretory cell differentiation. Considering the practical difficulties, it is not necessary to be included in the current manuscript. However, it shall be a feasible biological experiment for future studies.

Reviewer #4 (Remarks to the Author):

The authors adequately addressed all prior concerns, no additional comments.